# Multi-task Causal Learning with Gaussian Processes

**Virginia Aglietti**
University of Warwick
The Alan Turing Institute
V.Aglietti@warwick.ac.uk

**Theodoros Damoulas**
University of Warwick
The Alan Turing Institute
T.Damoulas@warwick.ac.uk

**Mauricio A. Álvarez**
University of Sheffield
Mauricio.Alvarez@sheffield.ac.uk

**Javier González**
Microsoft Research Cambridge
Gonzalez.Javier@microsoft.com

## Abstract

This paper studies the problem of learning the correlation structure of a set of intervention functions defined on the directed acyclic graph (DAG) of a causal model. This is useful when we are interested in jointly learning the causal effects of interventions on different subsets of variables in a DAG, which is common in field such as healthcare or operations research. We propose the first multi-task causal Gaussian process (GP) model, which we call DAG-GP, that allows for information sharing *across* continuous interventions and *across* experiments on different variables. DAG-GP accommodates different assumptions in terms of data availability and captures the correlation between functions lying in input spaces of different dimensionality via a well-defined integral operator. We give theoretical results detailing *when* and *how* the DAG-GP model can be formulated depending on the DAG. We test both the quality of its predictions and its calibrated uncertainties. Compared to single-task models, DAG-GP achieves the best fitting performance in a variety of real and synthetic settings. In addition, it helps to select optimal interventions faster than competing approaches when used within sequential decision making frameworks, like active learning or Bayesian optimization.

## 1 Introduction

Solving decision making problems in a variety of domains such as healthcare, systems biology or operations research, often requires experimentation. By performing interventions one can understand how a system behaves when an action is taken and thus infer the cause-effect relationships of a phenomenon. Experiments are especially useful when observational causal inference methods do not provide accurate estimation of the causal effects. For instance, in healthcare, drugs are tested in randomized clinical trials before commercialization. Biologists might want to understand how genes interact in a cell once one of them is knocked out. Finally, engineers investigate the impact of design changes on complex physical systems by conducting experiments on digital twins [34]. Experiments in these scenarios are usually expensive, time-consuming, and, especially for field experiments, they may present ethical issues. Therefore, researchers generally have to trade-off cost, time, and other practical considerations to decide which experiments to conduct, if any, to learn about the system.

Consider the causal graph in Fig. 1 which describes how crop yield $Y$ is affected by soil fumigants $X$ and the level of eel-worm population at different times $\mathbf{Z} = \{Z_1, Z_2, Z_3\}$ [11, 27]. By performing a set of experiments, the investigator aims at learning the *intervention functions* relating the expected crop yield to each possible intervention set and level. Naïvely, one could achieve that by modelling each intervention function separately. However, this approach would disregard the correlation structure existing across experimental outputs and would increase the computational

complexity of the problem. Indeed, the intervention functions are correlated and each experiment carries information about the yield we would obtain by performing alternative interventions in the graph. For instance, observing the yield when running an experiment on the *intervention set* $\{X, Z_1\}$ and setting the value to the *intervention value* $\{x, z_1\}$, provides information about the yield we would get from intervening only on $X$ or on $\{X, Z_1, Z_2, Z_3\}$. This paper studies how to jointly model such intervention functions so as to transfer knowledge across different experimental setups and integrate observational and interventional data. The model proposed here enables proper uncertainty quantification of the causal effects thus allowing the definition of optimal experimental design strategies.

## 1.1 Motivation and Contributions

The framework proposed in this work combines causal inference with multi-task learning via Gaussian processes (GP, [30]). Probabilistic causal models are commonly used in disciplines where explicit experimentation may be difficult and the *do*-calculus [27] enables prediction of the effect of an intervention without performing the experiment. In *do*-calculus, different intervention functions are modelled individually and there is no information shared across experiments. Modelling the correlation across experiments is crucial especially when the number of observational data points is limited and experiments on some variables cannot be performed. Multi-task GP methods have been extensively used to model non-trivial correlations between outputs [4]. However, to the best of our knowledge, this is the first work focusing on intervention functions, possibly of different dimensionality, defined on a causal graph. Particularly, we make the following contributions:

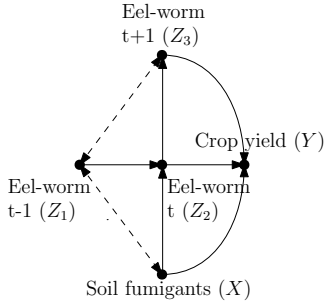

Figure 1: DAG for the crop yield. Nodes denote variables, arrows represent causal effects and dashed edges indicate unobserved confounders.

- We give theoretical results detailing *when* and *how* a causal multi-task model for the experimental outputs can be developed depending on the topology of the DAG of a causal model.

- Exploiting our theoretical results, we develop a joint probabilistic model for all intervention functions, henceforth named DAG-GP, which flexibly accommodates different assumptions in terms of data availability – both observational and interventional.

- We demonstrate how DAG-GP achieves the best fitting performance in a variety of experimental settings while enabling proper uncertainty quantification and thus optimal decision making when used within Active Learning (AL) and Bayesian Optimization (BO).

## 1.2 Related work

While there exists an extensive literature on multi-task learning with GPs [9, 4] and causality [28, 17], the literature on causal multi-task learning is very limited. In the causality literature, studies have focused on observational causal inference and have focused on the problem of transferring the causal effect of one given variable *across* environments [29, 6–8]. Several works have focused on domain adaptation problems [31, 26, 35] where data for a source domain is given, and the task is to predict the distribution of a target variable in a target domain. Closer to our work, [2] have developed a linear coregionalization model for learning the individual treatment effects via observational data. While [2] is the first paper conceptualizing causal inference as a multi-task learning problem, its focus is on modelling the correlation across intervention levels for a single intervention function corresponding to a dichotomous intervention variable. Finally, [24] studied the problem of identification of the causal effect of one intervention set in terms of available observational and experimental distributions. Focusing on the problem of identification, [24] does not provide a model for these distributions nor focuses on transfer across interventions. While one could repeat their procedure to get an expression for all possible intervention sets in the causal graph, this would not allow expressing all causal effects via a shared interventional distribution which is the focus of this paper. In addition, this paper focuses on settings where all causal effects are identifiable and interventional data are available. In these settings, the procedure in [24] would simplify and would either return the experimental output values, when these are available, or compute the causal effects via do-calculus.

Differently from these previous works, this paper focuses on transfer *within* a single environment, *across* experiments and *across* intervention levels. The set of functions we wish to learn have continuous input spaces of different dimensionality. Therefore, capturing their correlation requires placing a probabilistic model over the inputs which enables mapping between input spaces. The DAG, which we assumed to be known and is not available in standard multi-task settings, allows us to define such a model. Therefore, *existing multi-output* GP *models are not applicable to our problem.*

Our work is also related to the literature on causal decision making. Studies in this field have focused on multi-armed bandit problems [5, 21, 25, 22] and reinforcement learning [10, 14] settings where arms or actions correspond to interventions on a DAG. More recently, [1] proposed a Causal Bayesian Optimization (CBO) framework solving the problem of finding an optimal intervention in a DAG by modelling the intervention functions with GPs. Interestingly, the authors proposed a GP prior constructions, the so called *causal prior*, enabling the integration of observational and interventional data. More importantly, in CBO each function is modelled independently and their correlation is not accounted for when exploring the intervention space. This paper overcomes this limitation by introducing a multi-task model for experimental outputs. Finally, in the causal literature there has been a growing interest for experimental design algorithms to learn causal graphs [19, 18, 16] or the observational distributions in a graph [32]. Here we use our multi-task model within an AL framework so as to efficiently learn the experimental outputs in a causal graph.

## 2 Background and Problem setup

Consider a probabilistic structural causal model (SCM) [28] consisting of a directed acyclic graph $\mathcal{G}$ (DAG) and a four-tuple $\langle \mathbf{U}, \mathbf{V}, F, P(\mathbf{U}) \rangle$, where $\mathbf{U}$ is a set of independent *exogenous* background variables distributed according to the probability distribution $P(\mathbf{U})$, $\mathbf{V}$ is a set of observed *endogenous* variables and $F = \{f_1, \ldots, f_{|\mathbf{V}|}\}$ is a set of functions such that $v_i = f_i(\mathrm{Pa}_i, u_i)$ with $\mathrm{Pa}_i = \mathrm{Pa}(V_i)$ denoting the parents of $V_i$. $\mathcal{G}$ [1] encodes our knowledge of the existing causal mechanisms among $\mathbf{V}$. Within $\mathbf{V}$, we distinguish between two different types of variables: treatment variables $\mathbf{X}$ that can be manipulated and set to specific values[2] and output variables $\mathbf{Y}$ that represent the agent's outcomes of interest. Given $\mathcal{G}$, we denote the *interventional distribution* for two disjoint sets in $\mathbf{V}$, say $\mathbf{X}$ and $\mathbf{Y}$, as $P(\mathbf{Y}|\mathrm{do}(\mathbf{X} = \mathbf{x}))$. This is the distribution of $\mathbf{Y}$ obtained by intervening on $\mathbf{X}$ and fixing its value to $\mathbf{x}$ in the data generating mechanism, irrespective of the values of its parents. The interventional distribution differs from the *observational distribution* which is denoted by $P(\mathbf{Y}|\mathbf{X} = \mathbf{x})$. Under some identifiability conditions [15], *do*-calculus allows the estimation of interventional distributions and thus causal effects from observational distributions [27]. In this paper, we assume the causal effect for $\mathbf{X}$ on $\mathbf{Y}$ to be identifiable $\forall \mathbf{X} \in \mathcal{P}(\mathbf{X})$ with $\mathcal{P}(\mathbf{X})$ denoting the power set of $\mathbf{X}$.

### 2.1 Problem setup

Consider a DAG $\mathcal{G}$ and the related SCM. Define the set of intervention functions for $Y$ in $\mathcal{G}$ as:

$$\mathbf{T} = \{t_s(\mathbf{x})\}_{s=1}^{|\mathcal{P}(\mathbf{X})|} \qquad t_s(\mathbf{x}) = \mathbb{E}_{p(Y|\mathrm{do}(\mathbf{X}_s = \mathbf{x}))}[Y] = \mathbb{E}[Y|\mathrm{do}(\mathbf{X}_s = \mathbf{x})]. \tag{1}$$

with $\mathbf{X}_s \in \mathcal{P}(\mathbf{X})$ where $\mathcal{P}(\mathbf{X})$ is the power set of $\mathbf{X}$ minus the empty set[3] and $\mathbf{x} \in D(\mathbf{X}_s)$ where $D(\mathbf{X}_s) = \times_{X \in \mathbf{X}_s} D(X)$ with $D(X)$ denoting the *interventional domain* of $X$. Let $\mathcal{D}^O = \{\mathbf{x}_n, y_n\}_{n=1}^{N}$, with $\mathbf{x}_n \in \mathbb{R}^{|\mathbf{X}|}$ and $y_n \in \mathbb{R}$, be an observational dataset of size $N$ from this SCM. Consider an interventional dataset $\mathcal{D}^I = (\mathbf{X}^I, \mathbf{Y}^I)$ with $\mathbf{X}^I = \bigcup_s \{\mathbf{x}_{si}^I\}_{i=1}^{N_s^I}$ and $\mathbf{Y}^I = \bigcup_s \{y_{si}^I\}_{i=1}^{N_s^I}$ denoting the intervention levels and the function values observed from previously run experiments across sets in $\mathcal{P}(\mathbf{X})$. $N_s^I$ represents the number of experimental outputs observed for the intervention set $\mathbf{X}_s$. Our goal is to define a joint prior distribution $p(\mathbf{T})$ and compute the posterior $p(\mathbf{T}|\mathcal{D}^I)$ so as to make probabilistic predictions for $\mathbf{T}$ at some unobserved intervention sets and levels.

# 3 Multi-task learning of intervention functions

In this section we address the following question: *can we develop a joint model for the functions* $\mathbf{T}$ *in a causal graph and thus transfer information across experiments?*

To answer this question we study the correlation among functions in $\mathbf{T}$ which varies with the topology of $\mathcal{G}$. Inspired by previous works on latent force models [3], we show how any functions in $\mathbf{T}$ can be written as an integral transformation of some base function $f$, also defined starting from $\mathcal{G}$, via some integral operator $L_s$ such that $t_s(\mathbf{x}) = L_s(f)(\mathbf{x}), \forall \mathbf{X}_s \in \mathcal{P}(\mathbf{X})$. We first characterize the latent structure among experimental outputs and provide an explicit expression for both $f$ and $L_s$ for each intervention set (§3.1). Based on the properties of $\mathcal{G}$, we clarify when this function exists. Exploiting these results, we detail a new model to learn $\mathbf{T}$ which we call the DAG-GP model (§3.2). In DAG-GP we place a GP prior on $f$ and propagate our prior assumptions on the remaining part of the graph to analytically derive a joint distribution of the elements in $\mathbf{T}$. The resulting prior distribution incorporates the causal structure and enables the integration of observational and interventional data.

## 3.1 Characterization of the latent structure in a DAG

The following results provide a theoretical foundation for the multi-task causal GP model introduced later. In particular, they characterize when $f$ and $L_s$ exist and how to compute them thus fully characterizing when transfer across experiments is possible. All proofs are given in the appendix.

**Definition 3.1.** Consider a DAG $\mathcal{G}$ where the treatment variables are denoted by $\mathbf{X}$. Let $\mathbf{C}$ be the set of variables directly confounded with $Y$, $\mathbf{C}^N$ be the set of variables in $\mathbf{C}$ that are not colliders[4] and $\mathbf{I}$ be the set $\mathrm{Pa}(Y)$. For each $\mathbf{X}_s \in \mathcal{P}(\mathbf{X})$ we define the following sets:

- $\mathbf{I}_s^N = \mathbf{I} \backslash (\mathbf{X}_s \cap \mathbf{I})$ represents the set of variables in $\mathbf{I}$ not included in $\mathbf{X}_s$.

- $\mathbf{C}_s^I = \mathbf{C}^N \cap \mathbf{X}_s$ is the set of variables in $\mathbf{C}$ which are included in $\mathbf{X}_s$ and are not colliders.

- $\mathbf{C}_s^N = \mathbf{C}^N \backslash \mathbf{C}_s^I$ is the set of variables in $\mathbf{C}$ that are neither included in $\mathbf{X}_s$ nor colliders.

In the following theorem $\mathbf{v}_s^N$ gives the values for the variables in the set $\mathbf{I}_s^N$ while $\mathbf{c}$ represents the values for the set $\mathbf{C}^N$ which are partition in $\mathbf{c}_s^N$ and $\mathbf{c}_s^I$ depending on the set $\mathbf{X}_s$ we are considering.

**Theorem 3.1. Causal operator.** *Consider a causal graph $\mathcal{G}$ and a related* SCM *where the output variable and the treatment variables are denoted by $Y$ and $\mathbf{X}$ respectively. Denote by $\mathbf{C}$ the set of variables in $\mathcal{G}$ that are directly confounded with $Y$ and let $\mathbf{I}$ be the set $\mathrm{Pa}(Y)$. Assume that $\mathbf{C}$ does not include nodes that have both unconfounded incoming and outcoming edges. It is possible to prove that, $\forall \mathbf{X}_s \in \mathcal{P}(\mathbf{X})$, the intervention function $t_s(\mathbf{x}) : D(\mathbf{X}_s) \rightarrow \mathbb{R}$ can be written as $t_s(\mathbf{x}) = L_s(f)(\mathbf{x})$ where*

$$L_s(f)(\mathbf{x}) = \int \cdots \int \pi_s(\mathbf{x}, (\mathbf{v}_s^N, \mathbf{c})) f(\mathbf{v}, \mathbf{c}) d\mathbf{v}_s^N d\mathbf{c}, \qquad (2)$$

*with $f(\mathbf{v}, \mathbf{c}) = \mathbb{E}\left[Y | do\left(\mathbf{I} = \mathbf{v}\right), \mathbf{C}^N = \mathbf{c}\right]$ representing a shared latent function and $\pi_s(\mathbf{x}, (\mathbf{v}_s^N, \mathbf{c})) = p(\mathbf{c}_s^I | \mathbf{c}_s^N) p(\mathbf{v}_s^N, \mathbf{c}_s^N | do\left(\mathbf{X}_s = \mathbf{x}\right))$ giving the integrating measure for the set $\mathbf{X}_s$.*

We call $L_s(f)(\mathbf{x})$ the *causal operator*, $(\mathbf{I} \cup \mathbf{C})$ the *base set*, $f(\mathbf{v}, \mathbf{c})$ the *base function* and $\pi_s(\cdot, \cdot)$ the *integrating measure* of the set $\mathbf{X}_s$. A simple limiting case arises when the DAG does not include variables directly confounded with $Y$ or $\mathbf{C}$ only includes colliders. In this case $\mathbf{C} = \varnothing$ and the base function is included in $\mathbf{T}$. Theorem 3.1 provides a mechanism to reconstruct all causal effects emerging from $\mathcal{P}(\mathbf{X})$ using the base function as a "driving force". In particular, the integrating measures can be seen as Green's functions incorporating the DAG structure [3].

**Examples of $\pi_s(\mathbf{x}, (\mathbf{v}_s^N, \mathbf{c}))$:** For the DAG in Fig. 4(a) the integrating measure for $t_X(x)$ is $\pi_s(\mathbf{x}, (\mathbf{v}_s^N, \mathbf{c})) = \pi_X(x, (\mathbf{v}_X^N, \mathbf{c})) = p(z | do\left(X = x\right))$ as $\mathbf{I}_X^N = Z \backslash (X \cap Z) = Z$, $\mathbf{C}_X^I = \varnothing$ and $\mathbf{C}_X^N = \varnothing$ thus $\mathbf{v}_s^N = z$ while $\mathbf{c}_s^I$ and $\mathbf{c}_s^N$ are not defined. Similar expressions can be derived for the DAG in Fig. 4(b). Focusing on the computation of $t_B(b) = \mathbb{E}_Y[do\left(B = b\right)]$ we have: $\mathbf{I}_B^N = \{E, D\} \backslash (B \cap \{E, D\}) = \{E, D\}$, $\mathbf{C}_B^I = B$ and $\mathbf{C}_B^N = A$. We can then write $\pi_s(\mathbf{x}, (\mathbf{v}_s^N, \mathbf{c})) = \pi_B(b, (\mathbf{v}_B^N, \mathbf{c})) = p(b' | a) p(d, e, a | do\left(B = b\right)) = p(b') p(d, e, a | do\left(B = b\right))$. Full details for all DAGs used in this paper are given in the appendix.

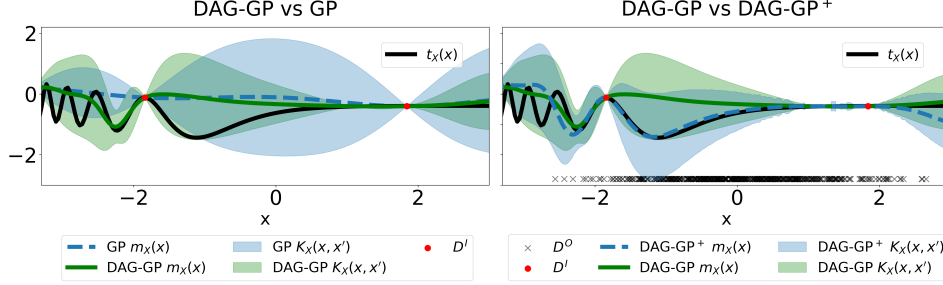

Figure 2: Posterior mean and variance for $t_X(\mathbf{x})$ in the DAG of Fig. 4 (a) (without the red edge). For both plots $m_X(\cdot)$ and $K_X(\cdot, \cdot)$ give the posterior mean and standard deviation respectively. *Left*: Comparison between the DAG-GP model and a single-task GP model (GP). DAG-GP captures the behaviour of $t_X(\mathbf{x})$ in areas where $\mathcal{D}^I$ is not available (see area around $x = -2$) while reducing the uncertainty via transfer due to available data for $\mathbf{z}$ (see appendix). *Right*: Comparison between DAG-GP with the causal prior (DAG-GP$^+$) and a standard prior with zero mean and RBF kernel (DAG-GP). In addition to transfer, DAG-GP$^+$ captures the behaviour of $t_X(\mathbf{x})$ in areas where $\mathcal{D}^O$ (black $\times$) is available (see region $[-2, 0]$) while inflating the uncertainty in areas with no observational data.

While these results can be further generalized to select $\mathbf{I}$ to be different from $\mathrm{Pa}(Y)$, this choice is particularly useful due to the following result.

**Corollary 3.1. Minimality of I.** *The smallest set $\mathbf{I}$ for which Eq.* (2) *holds is given by $\mathrm{Pa}(Y)$.*

The dimensionality of $\mathbf{I}$ when chosen as $\mathrm{Pa}(Y)$ has properties that have been previously studied in the literature. In the context of optimization [1], it corresponds to the so-called causal intrinsic dimensionality, which refers to the effective dimensionality of the space in which a function is optimized when causal information is available. The existence of $f$ depends on the properties of the nodes in $\mathbf{C}$ which also represents the smallest set for which Eq. (2) holds (§1.4 in the supplement).

**Theorem 3.2. Existence of $f$.** *If $\mathbf{C}$ includes nodes that have both unconfounded incoming and outcoming edges the function $f$ does not exist.*

When $f$ does not exist, full transfer across *all* functions in $\mathbf{T}$ is not possible (DAGs with red edges in Fig. 4). However, these results enable a model for partial transfer across a subset of $\mathbf{T}$ (§2 supp.).

## 3.2 The DAG-GP model

Next, we introduce the DAG GP model based on the results from the previous section.

**Model Likelihood:** Let $\mathcal{D}^I = (\mathbf{X}^I, \mathbf{Y}^I)$ be the interventional dataset defined in Section 2.1. Denote by $\mathbf{T}^I$ the collection of intervention vector-valued functions computed at $\mathbf{X}^I$. Each entry $y_{si}^I$ in $\mathbf{Y}^I$, is assumed to be a noisy observation of the corresponding function $t_s$ at $\mathbf{x}_i^I$:

$$y_{si}^I = t_s(\mathbf{x}_i^I) + \epsilon_{si}, \text{ for } s = 1, \dots, |\mathcal{P}(\mathbf{X})| \text{ and } i = 1, \dots, N_s^I, \tag{3}$$

with $\epsilon_{si} \sim \mathcal{N}(0, \sigma^2)$. In compact form, the joint likelihood function is $p(\mathbf{Y}^I | \mathbf{T}^I, \sigma^2) = \mathcal{N}(\mathbf{T}^I, \sigma^2 \mathbf{I})$.

**Prior distribution on T:** To define a join prior on the set of intervention functions, $p(\mathbf{T})$, we take the following steps. First, we follow [1] to place a *causal prior* on $f$, the base function of the DAG. Second, we propagate this prior on $f$ through all elements in $\mathbf{T}$ via the causal operator in Eq. (2).

*Step 1, causal prior on the base function*: The key idea of the causal prior, already used in [1], is to use the observational dataset $\mathcal{D}^O$ and the *do*-calculus to construct the prior mean and variance of a GP that is used to model an intervention function. Our aim is to compute such prior for the causal effect of the base set $\mathbf{I} \cup \mathbf{C}$ on $Y$. The causal prior has the benefit of carrying causal information but at the expense of requiring $\mathcal{D}^O$ to estimate the causal effect. Any sensible prior can be used in this step, so the availability of $\mathcal{D}^O$ is not strictly necessity. However, in this paper we stick to the causal prior since it provides an explicit way of combining experimental and observational data.

For simplicity we use $\mathbf{b} = (\mathbf{v}, \mathbf{c})$ to denote in compact form the values of the variables in the base set $\mathbf{I} = \mathbf{v}$ and $C = \mathbf{c}$. Using *do-calculus* we can compute $\hat{f}(\mathbf{b}) = \hat{f}(\mathbf{v}, \mathbf{c}) = \hat{\mathbb{E}}[Y | \mathrm{do}\,(\mathbf{I} = \mathbf{v}), \mathbf{c}]$ and

$\hat{\sigma}(\mathbf{b}) = \hat{\sigma}(\mathbf{v}, \mathbf{c}) = \hat{\mathbb{V}}[Y | \mathrm{do}\,(\mathbf{I} = \mathbf{v})\,, \mathbf{c}]^{1/2}$ where $\hat{\mathbb{V}}$ and $\hat{\mathbb{E}}$ represent the variance and expectation of the causal effects estimated from $\mathcal{D}^O$. The *causal prior* is defined as:

$$f(\mathbf{b}) \sim \mathcal{GP}(m(\mathbf{b}), K(\mathbf{b}, \mathbf{b}'))$$

$$m(\mathbf{b}) = \hat{f}(\mathbf{b})$$

$$K(\mathbf{b}, \mathbf{b}') = k_{\mathrm{RBF}}(\mathbf{b}, \mathbf{b}') + \hat{\sigma}(\mathbf{b})\hat{\sigma}(\mathbf{b}')$$

where $m(\mathbf{b})$ and $K(\mathbf{b}, \mathbf{b}')$ represents the prior mean and variance respectively. The term $k_{\mathrm{RBF}}(\mathbf{b}, \mathbf{b}') := \sigma_f^2 \exp(-||\mathbf{b} - \mathbf{b}'||^2/2l^2)$ denotes the radial basis function (RBF) kernel, which is added to provide more flexibility to the model.

*Step 2, propagating the distribution to all elements in* $\mathbf{T}$*:* In Section 3.1 we showed how, $\forall \mathbf{X}_s \in \mathcal{P}(\mathbf{X})$, $t_s(\mathbf{x}) = L_s(f)(\mathbf{x})$ with $f$ given by the intervention function defined in Theorem 3.1. By linearity of the causal operator, placing a GP prior on $f$ induces a well-defined joint GP prior distribution on $\mathbf{T}$. In particular, for each $\mathbf{X}_s \in \mathcal{P}(\mathbf{X})$, we have $t_s(\mathbf{x}) \sim \mathcal{GP}(m_s(\mathbf{x}), k_s(\mathbf{x}, \mathbf{x}'))$ with:

$$m_s(\mathbf{x}) = \int \cdots \int m(\mathbf{b})\pi_s\,(\mathbf{x}, \mathbf{b}_s)\,\mathrm{d}\mathbf{b}_s \tag{4}$$

$$k_s(\mathbf{x}, \mathbf{x}') = \int \cdots \int K(\mathbf{b}, \mathbf{b}')\pi_s\,(\mathbf{x}, \mathbf{b}_s)\,\pi_s\,(\mathbf{x}', \mathbf{b}'_s)\,\mathrm{d}\mathbf{b}_s\mathrm{d}\mathbf{b}'_s. \tag{5}$$

where $\mathbf{b}_s = (\mathbf{v}_s^N, \mathbf{c})$ is the subset of $\mathbf{b}$ including only the $\mathbf{v}$ values corresponding to the set $\mathbf{I}_s^N$.

Let $D$ be a finite set of inputs for the functions in $\mathbf{T}$, that is $D = \bigcup_s \{\mathbf{x}_{si}\}_{i=1}^M$. $\mathbf{T}$ computed in $D$ follows a multivariate Gaussian distribution that is $\mathbf{T}^D \sim \mathcal{N}(m_{\mathbf{T}}(D), K_{\mathbf{T}}(D, D))$ with $K_{\mathbf{T}}(D, D) = (K_{\mathbf{T}}(\mathbf{x}, \mathbf{x}'))_{\mathbf{x} \in D, \mathbf{x}' \in D}$ and $m_{\mathbf{T}}(D) = (m_{\mathbf{T}}(\mathbf{x}))_{\mathbf{x} \in D}$. In particular, for two generic data points $\mathbf{x}_{si}, \mathbf{x}_{s'j} \in D$ with $s$ and $s'$ denoting two *distinct* functions we have $m_{\mathbf{T}}(\mathbf{x}_{si}) = \mathbb{E}[t_s(\mathbf{x}_i)] = m_s(\mathbf{x}_i)$ and $K_{\mathbf{T}}(\mathbf{x}_{si}, \mathbf{x}_{s'j}) = \mathrm{Cov}[t_s(\mathbf{x}_i), t_{s'}(\mathbf{x}_j)]$.

When computing the covariance function across intervention sets and intervention levels we differentiate between two cases. When both $t_s$ and $t_{s'}$ are different from $f$, we have:

$$\mathrm{Cov}[t_s(\mathbf{x}_i), t_{s'}(\mathbf{x}_j)] = \int \cdots \int K(\mathbf{b}, \mathbf{b}')\pi_s\,(\mathbf{x}_i, \mathbf{b}_s)\,\pi_{s'}\,(\mathbf{x}_j, \mathbf{b}'_{s'})\,\mathrm{d}\mathbf{b}_s\mathrm{d}\mathbf{b}'_{s'}.$$

If one of the two functions equals $f$, this expression further reduces to:

$$\mathrm{Cov}[t_s(\mathbf{x}_i), t_{s'}(\mathbf{x}_j)] = \int K(\mathbf{b}, \mathbf{b}')\pi_{s'}\,(\mathbf{x}_j, \mathbf{b}'_{s'})\,\mathrm{d}\mathbf{b}'_{s'}.$$

Note that the integrating measures $\pi_s\,(\cdot, \cdot)$ and $\pi_{s'}\,(\cdot, \cdot)$ allow to compute the covariance between points that are defined on spaces on possibly different dimensionality, *a scenario that traditional multi-output* GP *models are unable to handle*. The prior $p(\mathbf{T})$ enables to merge different data types and to account for the natural correlation structure among interventions defined by the topology of the DAG. For this reason we call this formulation the DAG-GP model. The parameters in Eqs. (4)–(5) can be computed in closed form only when $K(\mathbf{b}, \mathbf{b}')$ is an RBF kernel and the integrating measures are assumed to be Gaussian distributions. In all other cases, one needs to resort to numerical approximations e.g. Monte Carlo integration in order to compute the parameters of each $t_s(\mathbf{x})$.

**Posterior distribution on T:** The posterior distribution $p(\mathbf{T}^D | \mathcal{D}^I)$ can be derived analytically via standard GP updates. For any set $D$, $p(\mathbf{T}^D | \mathcal{D}^I)$ will be Gaussian with parameters $m_{\mathbf{T} | \mathcal{D}^I}(D) = m_{\mathbf{T}}(D) + K_{\mathbf{T}}(D, \mathbf{X}^I)[K_{\mathbf{T}}(\mathbf{X}^I, \mathbf{X}^I) + \sigma^2 \mathbf{I}](\mathbf{T}^I - m_{\mathbf{T}}(\mathbf{X}^I))$ and $K_{\mathbf{T} | \mathcal{D}^I}(D, D) = K_{\mathbf{T}}(D, D) - K_{\mathbf{T}}(D, \mathbf{X}^I)[K_{\mathbf{T}}(\mathbf{X}^I, \mathbf{X}^I) + \sigma^2 \mathbf{I}]K_{\mathbf{T}}(\mathbf{X}^I, D)$. See Fig. 2 for an illustration of the DAG-GP model. The time complexity of the algorithm is $\mathcal{O}(N^3)$ with $N$ denoting the size of $\mathcal{D}^I$. This complexity can be reduced by resorting to sparse GP approximations e.g. inducing points approximations.

## 4   A helicopter view

Different variations of the DAG-GP model can be considered depending on the availability of both observational $\mathcal{D}^O$ and interventional data $\mathcal{D}^I$ (Fig. 3). Our goal here is not to be exhaustive, nor prescriptive, but to help to give some perspective. When $\mathcal{D}^I$ is not available do-calculus is the only way to learn $\mathbf{T}$, which in turns requires $\mathcal{D}^O$. When both data types are not available, learning $\mathbf{T}$ via a

| | | Interventional data | |
| | | **<span style="color:red">No</span>** | **<span style="color:green">Yes</span>** |
| | | Single-task | Multi-task |
| | | **GP** | **DAG-GP** |
| Observational data **<span style="color:red">No</span>** | Mechanistic model | $p(T) = \prod_s p(t_s(\mathbf{x}))$ | $p(\mathbf{T}) = \prod_s p(t_s(\mathbf{x})\|f)$ |
| | | $t_s(\mathbf{x}) \sim \mathcal{GP}(0, K_{RBF}(\mathbf{x},\mathbf{x}'))$ | $t_s(\mathbf{x}) = \int f(\mathbf{b})\pi_s(\mathbf{x},\mathbf{b}_s)\mathrm{d}\mathbf{b}_s$ |
| | | | $f(\mathbf{b}) \sim \mathcal{GP}(0, K_{RBF}(\mathbf{b},\mathbf{b}'))$ |
| | | **GP$^+$** | **DAG-GP$^+$** |
| **<span style="color:green">Yes</span>** | *do*-calculus | $p(T) = \prod_s p(t_s(\mathbf{x}))$ | $p(\mathbf{T}) = \prod_s p(t_s(\mathbf{x})\|f)$ |
| | | $t_s(\mathbf{x}) \sim \mathcal{GP}(m^+(\mathbf{x}), K^+(\mathbf{x},\mathbf{x}'))$ | $t_s(\mathbf{x}) = \int f(\mathbf{b})\pi_s(\mathbf{x},\mathbf{b}_s)\mathrm{d}\mathbf{b}_s$ |
| | | | $f(\mathbf{b}) \sim \mathcal{GP}(m^+(\mathbf{b}), K^+(\mathbf{b},\mathbf{b}'))$ |

Figure 3: Models for learning the intervention functions $\mathbf{T}$ defined on a DAG. The *do*-calculus allows estimating $\mathbf{T}$ when only the observational data is available. When the interventional data is also available, one can use a single-task model (denoted by GP) or a multi-task model (denoted by DAG-GP). When both data types are available one can combine them using the causal prior parameters represented by $m^+(\cdot)$ and $k^+(\cdot,\cdot)$. The resulting models are denoted by GP$^+$ and DAG-GP$^+$.

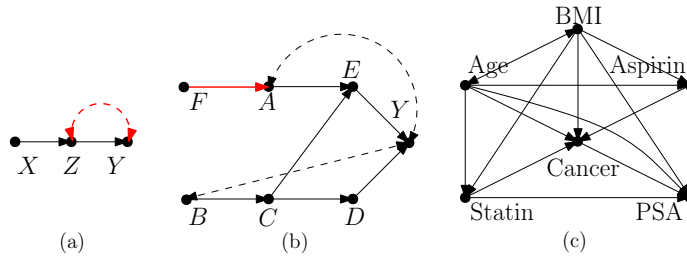

Figure 4: Examples of DAGs (in black) for which $f$ exists and the DAG-GP model can be formulated. The red edges, if added, prevent the identification of $f$ making the transfer via DAG-GP not possible.

probabilistic model is not possible unless the causal effects can be transported from an alternative population. In this case mechanistic models based on physical knowledge of the process under investigation are the only option. When $\mathcal{D}^I$ is available one can consider a single task or a multi-task model. If $f$ does not exist, a single GP model needs to be considered for each intervention function. Depending on the availability of $\mathcal{D}^O$, integrating observational data into the prior distribution (denoted by GP$^+$) or adopting a standard prior (denoted by GP) are the two alternatives. In both cases, the experimental information is not shared across functions and learning $\mathbf{T}$ requires intervening on all sets in $\mathcal{P}(\mathbf{X})$. When instead $f$ exists, DAG-GP can be used to transfer interventional information and, depending on $\mathcal{D}^O$, also incorporating observational information a priori (DAG-GP$^+$).

## 5 Experiments

This section evaluates the performance of the DAG-GP model on two synthetic settings and on a real world healthcare application (Fig. 4). We first learn $\mathbf{T}$ with fixed observational and interventional data (§5.1) and then use the DAG-GP model to solve active learning (AL) (§5.2) and Bayesian Optimization (BO) (§5.3)[5]. Implementation details are given in the supplement.

**Baselines:** We run our algorithm both with (DAG-GP$^+$) and without (DAG-GP) the causal prior and compare against the alternative models described in Fig. 3. Note that we do not compare against alternative multi-task GP models because, as mentioned in Section 1.2, the models existing in the literature cannot be straightforwardly applied to our problem. In addition, given that we assume full identifiability of causal effects and availability of $\mathcal{D}^I$, the mean results for GP$^+$ correspond to the results we would get by applying the gID procedure in [24] (see §1.2 for a discussion of this method).

**Performance measures:** We run all models with different initialisation of $\mathcal{D}^I$ and different sizes of $\mathcal{D}^O$. We report the root mean square error (RMSE) performances together with standard errors across replicates. For the AL experiments we show the RMSE evolution as the size of $\mathcal{D}^I$ increases. For the BO experiments we report the convergence performances to the global optimum.

Table 1: RMSE performances across 10 initializations of $\mathcal{D}^I$. See Fig. 3 for details on the compared methods. *do* stands for the *do*-calculus. $N$ is the size of $\mathcal{D}^O$. Standard errors in brackets.

| | $N = 30$ | | | | | $N = 100$ | | | | |
|---|---|---|---|---|---|---|---|---|---|---|
| | DAG-GP$^+$ | DAG-GP | GP$^+$ | GP | *do* | DAG-GP$^+$ | DAG-GP | GP$^+$ | GP | *do* |
| DAG1 | **0.46** | 0.57 | 0.60 | 0.77 | 0.70 | **0.43** | 0.57 | 0.45 | 0.77 | 0.52 |
| | (0.06) | (0.09) | (0.2) | (0.27) | - | (0.05) | (0.08) | (0.05) | (0.27) | - |
| DAG2 | **0.44** | 0.45 | 0.62 | 1.26 | 1.40 | **0.36** | 0.41 | 0.58 | 1.28 | 1.41 |
| | (0.1) | (0.13) | (0.10) | (0.11) | - | (0.09) | (0.12) | (0.07) | (0.11) | - |
| DAG3 | **0.05** | 0.44 | 0.23 | 0.89 | 0.18 | **0.06** | 0.44 | 0.48 | 0.89 | 0.23 |
| | (0.04) | (0.12) | (0.03) | (0.23) | - | (0.04) | (0.12) | (0.06) | (0.23) | - |

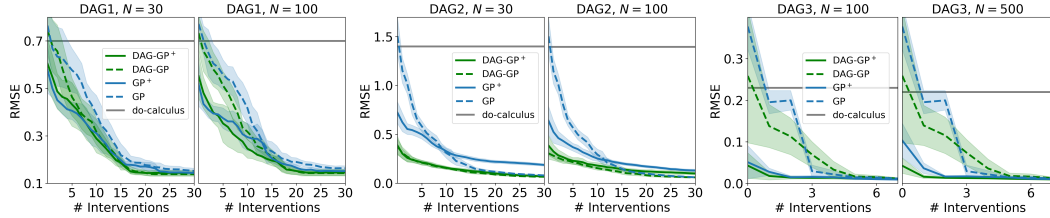

Figure 5: AL results. Convergence of the RMSE performance across functions in $\mathbf{T}$ and across replicates as more experiments are collected. DAG-GP$^+$ gives our algorithm with the causal prior while DAG-GP is our algorithm with a standard prior. $\#$ interventions is the number of experiments for each $\mathbf{X}_s$. Shaded areas give $\pm$ standard deviation. See Fig. 3 for details on the compared methods.

## 5.1 Learning T from data

We test the algorithm on the DAGs in Fig. 4 and refer to them as (a) DAG1, (b) DAG2 and (c) DAG3. DAG3 is taken from [33] and [13] and is used to model the causal effect of statin drugs on the levels of prostate specific antigen (PSA). We consider the nodes $\{A, C\}$ in DAG2 and $\{age, BMI, cancer\}$ in DAG3 to be non-manipulative. We set the size of $\mathcal{D}^I$ to $5 \times |\mathbf{T}|$ for DAG1 ($|\mathbf{T}| = 2$), to $3 \times |\mathbf{T}|$ for DAG2 ($|\mathbf{T}| = 6$) and to $|\mathbf{T}|$ for DAG3 ($|\mathbf{T}| = 3$). As expected, GP$^+$ outperforms GP incorporating the information in $\mathcal{D}^O$ (Tab. 1). Interestingly, GP$^+$ also outperforms DAG-GP in DAG3 when $N = 30$ and in DAG1 when $N = 100$. This depends on the effect that $\mathcal{D}^O$ has, through its size $N$ and its coverage of the interventional domains, on both the causal prior and the estimation of the integrating measures. Lower $N$ and coverage imply not only a less precise estimation of the *do*-calculus but also a worse estimation of the integrating measures and thus a lower transfer of information. Higher $N$ and coverage imply more accurate estimation of the causal prior parameters and enhanced transfer of information across experiments. In addition, the way $\mathcal{D}^O$ affects the performance results it's specific to the DAG structure and to the distribution of the exogenous variables in the SCM. More importantly, Tab. 1 shows how DAG-GP$^+$ consistently outperforms all competing methods by successfully integrating different data sources and transferring interventional information across functions in $\mathbf{T}$. Differently from competing methods, these results holds across different $N$ and $\mathcal{D}^I$ values making DAG-GP$^+$ a robust default choice for any application.

## 5.2 DAG-GP as surrogate model in Active Learning

The goal of AL is to design a sequence of function evaluations to perform in order to learn a target function as quickly as possible. Denote by $D$ a set of inputs for the functions in $\mathbf{T}$, that is $D = \bigcup_s D_s$ with $D_s \subset D(\mathbf{X}_s)$ and a set $A \subset D$ of size $k$. We would like to select $A$, that is select the both the functions to be observed and the locations, such that we maximize the reduction of entropy in the remaining unobserved locations:

$$A^\star = \underset{A:|A|=k}{\operatorname{argmax}} H(\mathbf{T}(D \backslash A)) - H(\mathbf{T}(D \backslash A)|\mathbf{T}(A)).$$

where $\mathbf{T}(D \backslash A)$ denotes the set of functions $\mathbf{T}$ evaluated in $D \backslash A$, $\mathbf{T}(D \backslash A)|\mathbf{T}(A)$ gives the distribution for $\mathbf{T}$ at $(D \backslash A)$ given that we have observed $\mathbf{T}(A)$ while $H(\cdot)$ represents the entropy. While this problem is NP-complete, Krause et al. [20] proposed an efficient greedy algorithm providing an approximation for $A$. This algorithm starts with $A = \varnothing$ and solves the problem sequentially by selecting, at every step $j$, a point $\mathbf{x}_{sj} = \operatorname{argmax}_{\mathbf{x}_{sj} \in D \backslash A} H(t_s(\mathbf{x})|A) - H(t_s(\mathbf{x})|D \backslash (A \cup \mathbf{x}_{sj}))$.

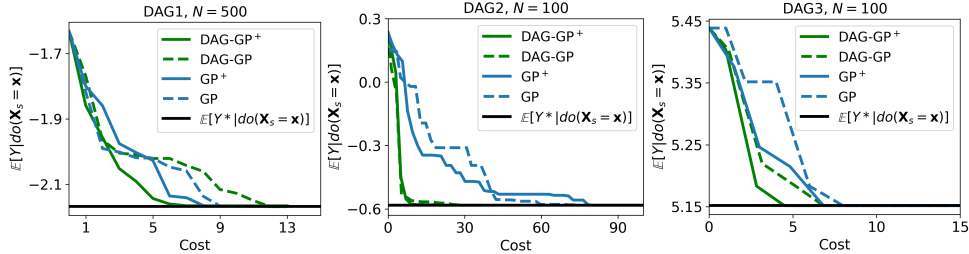

Figure 6: BO results. Convergence of the CBO algorithm to the global optimum ($\mathbb{E}[Y^\star|\mathrm{do}\,(\mathbf{X}_s = \mathbf{x})]$) when our algorithm is used as a surrogate model with (DAG-GP$^+$) and without (DAG-GP) the causal prior. See the supplement for standard deviations across replicates.

In order to select the next intervention level and intervention set, while properly accounting for uncertainty reduction, one can use the DAG-GP model for $\mathbf{T}$.

Fig. 5 shows the RMSE performances as more interventional data are collected and the DAG-GP model is used within the AL algorithm proposed by [20]. Across different $N$ settings, DAG-GP$^+$ converges to the lowest RMSE performance faster then competing methods by collecting evaluations in areas where: (i) $\mathcal{D}^O$ does not provide information and (ii) the predictive variance is not reduced by the experimental information transferred from the other interventions. As mentioned before, $\mathcal{D}^O$ impacts on the causal prior parameters via the *do*-calculus computations. When the latter are less precise, because of lower $N$ or lower coverage of the interventional domains, the model variances for DAG-GP$^+$ or GP$^+$ are inflated. Therefore, when DAG-GP$^+$ or GP$^+$ are used as surrogate models, the interventions are collected mainly in areas where $\mathcal{D}^O$ is not observed thus slowing down the exploration of the interventional domains and the convergence to the minimum RMSE (Fig. 5 DAG2, $N = 100$). See Section 5 in the supplement for more details about the use of the DAG-GP model within AL.

### 5.3 DAG-GP as surrogate model in Bayesian optimization

The goal of BO is to optimize a function which is costly to evaluate and for which an explicit functional form is not available by making a series of function evaluations. In a recent work, [1] introduced the CBO algorithm which finds the intervention optimising a target variable in a causal graph. In order to find the optimal intervention, CBO places a single-task GP model on all the intervention functions in a DAG. By modeling these functions independently, CBO does not account for their correlation when exploring the intervention space (see §6 in the supplement for more details). Replacing the independent surrogate models used by CBO with the DAG-GP framework significantly speed up the convergence to the global optimum. This is shown in Fig. 6 where the DAG-GP (with and without causal prior) is compared against the single-task models.

## 6 Conclusions

This paper addresses the problems of modelling the correlation structure of a set of intervention functions defined on the DAG of a causal model. We propose the DAG-GP model, which is based on a theoretical analysis of the DAG structure, and allows to share experimental information across interventions while integrating observational and interventional data via *do*-calculus. Our results demonstrate how DAG-GP outperforms competing approaches in term of fitting performances. In addition, our experiments show how integrating decision making algorithms with the DAG-GP model is crucial when designing optimal experiments as DAG-GP accounts for the uncertainty reduction obtained by transferring interventional data. Future work will extend the DAG-GP model to allow for transfer of experimental information *across* environments whose DAGs are partially different. In addition, we will focus on combining the proposed framework with a causal discovery algorithm so as to account for uncertainty in the graph structure.

## Broader Impact

Computing causal effects is an integral part of scientific inquiry, spanning a wide range of questions such as understanding behaviour in online systems, assessing the effect of social policies, or investigation the risk factors for diseases. By combining the theory of causality with machine learning

techniques, Causal Machine Learning algorithms have the potential to highly impact society and businesses by answering what-if questions, enabling policy-evaluation and allowing for data-driven decision making in real-world contexts. The algorithm proposed in this paper falls into this category and focuses on addressing causal questions in a fast and accurate way. As shows in the experiments, when used within decision making algorithms, the DAG-GP model has the potential to speed up the learning process and to enable optimal experimentation decisions by accounting for the multiple causal connections existing in the process under investigation and their cross-correlation. Our algorithm can be used by practitioners in several domains. For instance, it can be used to learn about the impact of environmental variables on coral calcification [12] or to analyse the effects of drugs on cancer antigens [13]. In terms of methodology, while the DAG-GP model represents a step towards a better model for automated decision making, it is based on the crucial assumption of knowing the causal graph. Learning the intervention functions of an incorrect causal graph might lead to incorrect inference and sub-optimal decisions. Therefore, more work needs to be done to account for the uncertainty in the graph structure.

## Acknowledgements

This work was supported by the EPSRC grant EP/L016710/1, The Alan Turing Institute under EPSRC grant EP/N510129/1 and the Lloyds Register Foundation programme on Data Centric Engineering. MAA has been financed by the EPSRC Research Projects EP/R034303/1 and EP/T00343X/1. MAA has also been supported by the Rosetrees Trust (ref: A2501).

## Footnotes

[1]As mentioned above, in this paper we assume $\mathcal{G}$ to be known. However, one could run a causal discovery algorithm as a pre-processing step or use interventional data to discriminate among graphs within the Markov equivalence class. We leave this for future research.

[2]This setting can be extended to include non-manipulative variables. See [23] for a definition of such nodes.

[3]We exclude the empty set as it corresponds to the observational distribution $t_{\varnothing}(\mathbf{x}) = \mathbb{E}[Y]$.

[4]Variables in $\mathbf{C}$ causally influenced by $\mathbf{X}$ and $Y$.

[5]Code and data for all the experiments is provided at `https://github.com/VirgiAgl/DAG-GP`.

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
