[Supplementary Material]

# Supplementary Material for "Multi-task Causal Learning with Gaussian Processes"

**Virginia Aglietti**
University of Warwick
The Alan Turing Institute
V.Aglietti@warwick.ac.uk

**Theodoros Damoulas**
University of Warwick
The Alan Turing Institute
T.Damoulas@warwick.ac.uk

**Mauricio A. Álvarez**
University of Sheffield
Mauricio.Alvarez@sheffield.ac.uk

**Javier González**
Microsoft Research Cambridge
Gonzalez.Javier@microsoft.com

## 1 Proofs of theorems and additional theoretical results

In this section we give the proofs for the theorems in the main text and an additional theoretical result regarding the minimality of the set $\mathbf{C}$.

### 1.1 Proof of Theorem 3.1

*Proof.* Consider a generic $\mathbf{X}_s \in \mathcal{P}(\mathbf{X})$. $\mathbf{v}_s^I$ and $\mathbf{v}_s^N$ denote the values for the sets $\mathbf{I}_s^I$ and $\mathbf{I}_s^N$ respectively. $\mathbf{c} = (\mathbf{c}_s^I \cup \mathbf{c}_s^N)$ represents the values for the set $\mathbf{C}^N$, $\mathbf{c}_s^N$ is the value of $\mathbf{C}_s^N$ and $\mathbf{c}_s^I$ gives the value for $\mathbf{C}_s^I$. Notice that we can write the intervention on $\mathbf{X}_s$, that is do $(\mathbf{X}_s = \mathbf{x})$, as do $\left(\mathbf{I}_s^I = \mathbf{v}_s^I\right) \cup$ do $\left(\mathbf{X}_s \backslash \mathbf{I}_s^I = \mathbf{x} \backslash \mathbf{v}_s^I\right)$. Any function $t_s(\mathbf{x}) \in \mathbf{T}$ can be written as:

$$
\begin{aligned}
t_s(\mathbf{x}) &= \mathbb{E}\left[Y | \mathrm{do}\left(\mathbf{X}_s = \mathbf{x}\right)\right] \\
&= \int \cdots \int \mathbb{E}\left[Y | \mathrm{do}\left(\mathbf{I}_s^I = \mathbf{v}_s^I\right), \mathrm{do}\left(\mathbf{X}_s \backslash \mathbf{I}_s^I = \mathbf{x} \backslash \mathbf{v}_s^I\right), \mathbf{I}_s^N = \mathbf{v}_s^N, \mathbf{C}_s^N = \mathbf{c}_s^N\right] \times \\
&\qquad p(\mathbf{v}_s^N, \mathbf{c}_s^N | \mathrm{do}\left(\mathbf{X}_s = \mathbf{x}\right)) \mathrm{d}\mathbf{v}_s^N \mathrm{d}\mathbf{c}_s^N
\end{aligned}
$$

$$
\begin{aligned}
&= \int \cdots \int \mathbb{E}\left[Y | \mathrm{do}\left(\mathbf{I}_s^I = \mathbf{v}_s^I\right), \mathrm{do}\left(\mathbf{X}_s \backslash \mathbf{I}_s^I = \mathbf{x} \backslash \mathbf{v}_s^I\right), \mathrm{do}\left(\mathbf{I}_s^N = \mathbf{v}_s^N\right), \mathbf{C}_s^N = \mathbf{c}_s^N\right] \times \\
&\qquad p(\mathbf{v}_s^N, \mathbf{c}_s^N | \mathrm{do}\left(\mathbf{X}_s = \mathbf{x}\right)) \mathrm{d}\mathbf{v}_s^N \mathrm{d}\mathbf{c}_s^N \qquad \text{by} \quad Y \perp\!\!\!\perp \mathbf{I}_s^N | \mathbf{X}_s, \mathbf{C}_s^N \text{ in } \mathcal{G}_{\overline{\mathbf{X}_s \underline{\mathbf{I}_s^N}}} \qquad (1)
\end{aligned}
$$

$$
\begin{aligned}
&= \int \cdots \int \mathbb{E}\left[Y | \mathrm{do}\left(\mathbf{I}_s^I = \mathbf{v}_s^I\right), \mathrm{do}\left(\mathbf{I}_s^N = \mathbf{v}_s^N\right), \mathbf{C}^N = \mathbf{c}_s^N\right] \times \\
&\qquad p(\mathbf{v}_s^N, \mathbf{c}_s^N | \mathrm{do}\left(\mathbf{X}_s = \mathbf{x}\right)) \mathrm{d}\mathbf{v}_s^N \mathrm{d}\mathbf{c}_s^N \qquad \text{by} \quad Y \perp\!\!\!\perp (\mathbf{X}_s \backslash \mathbf{I}_s^I) | \mathbf{I}, \mathbf{C}_s^N \text{ in } \mathcal{G}_{\overline{\mathbf{I}(\mathbf{X}_s \backslash \mathbf{I}_s^I)(\mathbf{C}_s^N)}} \\
&\hspace{12cm} (2)
\end{aligned}
$$

$$= \int \cdots \int \mathbb{E}\left[Y|\mathrm{do}\left(\mathbf{I}=\mathbf{v}\right), \mathbf{C}_s^N = \mathbf{c}_s^N\right] p(\mathbf{v}_s^N, \mathbf{c}_s^N|\mathrm{do}\left(\mathbf{X}_s = \mathbf{x}\right)) d\mathbf{v}_s^N d\mathbf{c}_s^N$$

$$= \int \cdots \int \mathbb{E}\left[Y|\mathrm{do}\left(\mathbf{I}=\mathbf{v}\right), \mathbf{C}_s^N = \mathbf{c}_s^N, \mathbf{C}_s^I = \mathbf{c}_s^I\right] \times$$
$$p(\mathbf{c}_s^I|\mathrm{do}\left(\mathbf{I}=\mathbf{v}\right), \mathbf{C}_s^N = \mathbf{c}_s^N) p(\mathbf{v}_s^N, \mathbf{c}_s^N|\mathrm{do}\left(\mathbf{X}_s = \mathbf{x}\right)) d\mathbf{v}_s^N d\mathbf{c}_s^N d\mathbf{c}_s^I$$

$$= \int \cdots \int \mathbb{E}\left[Y|\mathrm{do}\left(\mathbf{I}=\mathbf{v}\right), \mathbf{C}^N = \mathbf{c}\right] p(\mathbf{c}_s^I|\mathbf{C}_s^N = \mathbf{c}_s^N) p(\mathbf{v}_s^N, \mathbf{c}_s^N|\mathrm{do}\left(\mathbf{X}_s = \mathbf{x}\right)) d\mathbf{v}_s^N d\mathbf{c}_s^N d\mathbf{c}_s^I$$
$$\tag{3}$$

$$\text{by} \quad \mathbf{C}_s^I \perp\!\!\!\perp \mathbf{I}|\mathbf{C}_s^N \quad \text{in} \quad \mathcal{G}_{\overline{\mathbf{I}}}$$
$$= \int \cdots \int \mathbb{E}\left[Y|\mathrm{do}\left(\mathbf{I}=\mathbf{v}\right), \mathbf{C}^N = \mathbf{c}\right] p(\mathbf{c}_s^I|\mathbf{c}_s^N) p(\mathbf{v}_s^N, \mathbf{c}_s^N|\mathrm{do}\left(\mathbf{X}_s = \mathbf{x}\right)) d\mathbf{v}_s^N d\mathbf{c}$$

$$= \int \cdots \int f(\mathbf{v}, \mathbf{c}) p(\mathbf{c}_s^I|\mathbf{c}_s^N) p(\mathbf{v}_s^N, \mathbf{c}_s^N|\mathrm{do}\left(\mathbf{X}_s = \mathbf{x}\right)) d\mathbf{v}_s^N d\mathbf{c}$$
$$\tag{4}$$

where Eq. (1) follows from Rule 2 of *do*-calculus while Eq. (2) and Eq. (3) follow from Rule 3 of *do*-calculus [3]. Eq. (4) gives the causal operator. ☐

### 1.2 Proof of Corollary 3.1

*Proof.* Suppose there exists another set $\mathbf{A}$, different from $\mathrm{Pa}(Y)$ and defined as $\mathbf{A} = \mathrm{Pa}(Y)\backslash\mathrm{Pa}(Y)_i$, where $\mathrm{Pa}(Y)_i$ represents a single variable in $\mathrm{Pa}(Y)$, such that Eq. (2) holds for every set $\mathbf{X}_s$. This means that $\mathbf{A}$ blocks the front-door paths from all $\mathbf{X}_s \in \mathcal{P}(\mathbf{X})$ to $Y$. That is, $\mathbf{A}$ also blocks the directed path from $\mathrm{Pa}(Y) \in \mathcal{P}(\mathbf{X})$ to $Y$ thus including descendants of $\mathrm{Pa}(Y)$ which are ancestors of $Y$. This contradicts the definition of a parent as a variable connected to $Y$ through a direct arrow. The same reasoning hold for every set non containing all parents of $Y$ thus $\mathrm{Pa}(Y)$ is the smallest set such that Eq. (2) holds. ☐

### 1.3 Proof of Theorem 3.2

*Proof.* Suppose that $\mathbf{C}$ includes a node, say $C_i$, that has both an incoming and an outcoming unconfounded edge. The unconfounded incoming edge implies the existence of a set $\mathbf{X}_s$ for which $C_i$ is a collider on the confounded path from $\mathbf{X}_s$ to $Y$. At the same time, the unconfounded outcoming edge implies the existence of a set $\mathbf{X}_{s'}$ such that $C_i$ is an ancestor that we need to condition on in order to clock the back-door paths from $\mathbf{X}_{s'}$ to $Y$. Consequently, the conditions $Y \perp\!\!\!\perp \mathbf{I}_s^N|\mathbf{X}_s, \mathbf{C}_s^N$ in $\mathcal{G}_{\overline{\mathbf{X}_s\mathbf{I}_s^N}}$ and $Y \perp\!\!\!\perp (\mathbf{X}_s\backslash\mathbf{I}_s^I)|\mathbf{I}, \mathbf{C}_s^N$ in $\mathcal{G}_{\overline{\mathbf{I}(\mathbf{X}_s\backslash\mathbf{I}_s^I)(\mathbf{C}_s^N)}}$ in Theorem 3.1 cannot hold, at the same time, for both $\mathbf{X}_s$ and $\mathbf{X}_{s'}$. Indeed, these independence conditions would be verified for $X_s$ when excluding $C_i$ from $\mathbf{C}^N$ while they would be verified for $X_{s'}$ when $C_i$ is included in $\mathbf{C}^N$. The same reasoning hold for every node in $\mathbf{C}$ having both incoming and outcoming unconfounded edges. Therefore, if $\mathcal{G}$ has one of such node, it is not possible to find a set $\mathbf{C}$ such that Eq. (2) holds from all $\mathbf{X}_s \in \mathcal{P}(\mathbf{X})$. ☐

### 1.4 Additional corollary

**Corollary 1.1.** The set $\mathbf{C}$ represents the smallest set for which Eq. (2) holds.

*Proof.* Suppose there exists another set $\mathbf{A}$, different from $\mathbf{C}$ and defined as $\mathbf{A} = \mathbf{C}\backslash C_i$ where $C_i \in \mathcal{P}(\mathbf{X})$ denotes a single variable in $\mathbf{C}$ that is not a collider. The set $\mathbf{A}$ need to be such that $Y \perp\!\!\!\perp (\mathbf{X}_s\backslash\mathbf{I}_s^I)|\mathbf{I}, \mathbf{A}_s^N$ in $\mathcal{G}_{\overline{\mathbf{I}(\mathbf{X}_s\backslash\mathbf{I}_s^I)(\mathbf{A}_s^N)}}$ $\forall \mathbf{X}_s$ in $\mathcal{P}(\mathbf{X})$. Consider $\mathbf{X}_s = C_i$ and notice that the back door path from $C_i$ to $Y$ is *not* blocked by conditioning on $\mathbf{I}$ or $\mathbf{A}_s^N$. Therefore $Y \not\perp\!\!\!\perp (\mathbf{X}_s\backslash\mathbf{I}_s^I)|\mathbf{I}, \mathbf{A}_s^N$ in $\mathcal{G}_{\overline{\mathbf{I}(\mathbf{X}_s\backslash\mathbf{I}_s^I)(\mathbf{A}_s^N)}}$ and $\mathbf{A}$ is not a valid set. The same reasoning holds for every set not containing all confounders of $Y$ thus $\mathbf{C}$ is the minimal set for $\mathbf{C}$. ☐

## 2 Partial transfer

The conditions in Theorem 3.1 allow for full transfer across *all* intervention functions in $\mathbf{T}$. As shown (see Theorem 3.2), this might not be possible when a subset $\mathbf{C}' \subset \mathbf{C}$ includes nodes directly confounded with $Y$ and with both unconfounded incoming and outcoming edges. However, we might still be interested in transferring information across a subset $\mathbf{T}' \subset \mathbf{T}$ which includes functions defined on $\mathcal{P}(\mathbf{X})' \subset \mathcal{P}(\mathbf{X})$. $\mathcal{P}(\mathbf{X})'$ is defined by excluding from $\mathcal{P}(\mathbf{X})$ those intervention sets including variables that have outcoming edges pointing into $\mathbf{C}'$ making the conditions in Theorem 3.1 satisfied for all sets in $\mathcal{P}(\mathbf{X})'$. For instance, consider Fig. 1 (b) with the red edge where $A$ is a confounded node that has both unconfounded incoming and outcoming edges. To block the path $E \leftarrow A \dashleftarrow\dashrightarrow Y$ we need to condition on $A$. However, conditioning on $A$ opens the path $F \rightarrow A \dashleftarrow\dashrightarrow Y$ making it impossible to define a base function. We can thus focus on a subset $\mathbf{T}'$ in which all functions including $\mathbf{C}' = \{A\}$ as an intervention variable have been excluded. This is equivalent to doing full transfer in Fig. 1 (b) with no incoming red edge in $A$.

## 3 Advantages of using the Causal operator

The causal operator allows us to write any $t_s(\mathbf{x})$ as an integral transformation of $f$. The integrating measure, which differ across $\mathbf{X}_s$, captures the dependency structure between the base set and the intervention set and can be reduced to *do*-free operations via *do*-calculus. Notice how, given our identifiability assumptions, all functions in $\mathbf{T}$ can also be computed by simply applying the rules of *do*-calculus when observational data are available. However, writing the functions via $L_s(f)(\mathbf{x})$ has several advantages:

- it allows to identify the correlation structure across functions and thus to specify a multi-task probabilistic model and share experimental information;
- it allows to learn those intervention functions for which we cannot run experiments via transfer;
- it allows to efficiently learn the set $\mathbf{T}$ when $\mathcal{P}(\mathbf{X})$ is large.

This is crucial when have limited observational data or we cannot run experiments on some intervention sets or the cardinality of $\mathcal{P}(\mathbf{X})$ is large. In the last case, specifying a model for each individual intervention function would not only be computationally expensive but might also lead to inconsistent prior specification across functions. Through the causal operator we can model a system by only making one single assumption on $f$ which is then propagated in the causal graph. When an intervention is performed, the information is propagated in the graph through the base function which links the different interventional functions. Using $f$ we avoid the specification of the correlation structure across every pair of intervention functions which would result in a combinatorial problem.

## 4 Single-task models for intervention functions

With single-task model we refer to the idea of placing an individual probabilistic model on the intervention function corresponding to each set in $\mathcal{P}(\mathbf{X})$. For each $\mathbf{X}_s \in \mathcal{P}(\mathbf{X})$ we have:

$$t_s(\mathbf{x}) \sim \mathcal{GP}(m(\mathbf{x}), K(\mathbf{x}, \mathbf{x}'))$$

Depending on the availability of $\mathcal{D}^O$, one can decide to set the prior parameters to standard value, e.g. $m(\mathbf{x}) = 0$ and $K(\mathbf{x}, \mathbf{x}') = K_{\text{RBF}}(\mathbf{x}, \mathbf{x}')$ or adopt the causal prior construction introduced by [1]. In both cases, the experimental information is not shared across functions and learning $\mathbf{T}$ requires intervening on all sets in $\mathcal{P}(\mathbf{X})$.

## 5 Active learning algorithm

Denote by $D$ a set of inputs for the functions in $\mathbf{T}$, that is $D = \bigcup_s D_s$ with $D_s \subset D(\mathbf{X}_s)$ and consider a subset $A \subset D$ of size $k$. We would like to select $A$, that is select the both the functions to be observed and the locations, such that we maximize the reduction of entropy in the remaining unobserved locations:

$$A^\star = \underset{A:|A|=k}{\text{argmax}} H(\mathbf{T}(D \backslash A)) - H(\mathbf{T}(D \backslash A)|\mathbf{T}(A)).$$

Figure 1: Snapshot of the AL algorithm for $t_Z(z)$ of the DAG in Fig. 1 (a) when $N^I = 8$ and $N = 100$. DAG-GP$^+$ is our algorithm with the causal GP prior. GP$^+$ is a single-task model with the same prior (see Fig. 3 for details on the compared models). Coloured crosses denote collected interventions while the red dot gives the common initial design.

where $\mathbf{T}(D\backslash A)$ denotes the set of functions $\mathbf{T}$ evaluated in $D\backslash A$, $\mathbf{T}(D\backslash A)|\mathbf{T}(A)$ gives the distribution for $\mathbf{T}$ at $(D\backslash A)$ given that we have observed $\mathbf{T}(A)$ while $H(\cdot)$ represents the entropy. This problem is NP-complete, Krause et al. [2] proposed an efficient greedy algorithm providing an approximation for $A$. This algorithm starts with an empty set $A = \varnothing$ and solves the problem sequentially by selecting, at every step $j$, a point $\mathbf{x}_{sj} = \text{argmax}_{\mathbf{x}_{sj} \in D\backslash A} H(t_s(\mathbf{x})|A) - H(t_s(\mathbf{x})|D\backslash(A \cup \mathbf{x}_{sj}))$. Both $H(t_s(\mathbf{x})|A) = \frac{1}{2}\log(2\pi\sigma^2_{\mathbf{x}_{sj}|A})$ and $H(t_s(\mathbf{x})|D\backslash(A \cup \mathbf{x}_{sj})) = \frac{1}{2}\log(2\pi\sigma^2_{\mathbf{x}_{sj}|D\backslash(A\cup\mathbf{x}_{sj})})$ do not depend on the observed $\mathbf{T}$ values thus the set $A$ can be selected before any function evaluation is collected.

In order to select the next intervention level and intervention set, while property accounting for uncertainty reduction, one can use the DAG-GP$^+$ model for $\mathbf{T}$. In this case, for every $\mathbf{X}_s$, both $\sigma^2_{\mathbf{x}_{sj}|A}$ and $\sigma^2_{\mathbf{x}_{sj}|D\backslash(A\cup\mathbf{x}_{sj})}$, which correspond to the variance terms of the kernel on $\mathbf{T}$, are determined by both the observational and the interventional data across all experiments. Fig. 1 shows a snapshot of the state of the AL algorithm for the toy example of Fig. 4 (a) when 8 interventional data points have been collected for $t_Z(z)$ and DAG-GP$^+$ is used. Both GP$^+$ and DAG-GP$^+$ avoid collecting data points in areas where the causal GP prior is already providing information thus making the model posterior mean equal to the true function (see region between $[0, 5]$). GP$^+$ is spreading the function evaluations on the remaining part of the input space collecting data points in the region $[5, 14]$. On the contrary, DAG-GP$^+$ drives the data points to be collected where neither observational nor interventional information can be transferred for the remaining tasks thus focusing on the border of the input space (see region $[14, 20]$). Combining an AL framework with DAG-GP$^+$ is thus crucial when designing optimal experiments as it allows to account for the uncertainty reduction obtained by transferring interventional data.

## 6 Bayesian Optimization

The goal of BO is to optimize a function which is costly to evaluate and for which an explicit functional form is not available by making a series of function evaluations. In a recent work, [1] introduce the CBO algorithm which solves the problem of finding an optimal intervention in a DAG. CBO optimizes a target node by accounting for the causal relationship between the inputs and placing a single-task GP model on the intervention functions. By modelling these functions independently, CBO does not account for their correlation when exploring the intervention space. For each $\mathbf{X}_s \in \mathcal{P}(\mathbf{X})$ we have:

$$t_s(\mathbf{x}) = \mathbb{E}[Y|\text{do}(\mathbf{X}_s = \mathbf{x})] \tag{5}$$

$$t_s(\mathbf{x}) \sim \mathcal{GP}(m^+(\mathbf{x}), K^+(\mathbf{x}, \mathbf{x}')) \tag{6}$$

where $m^+(\mathbf{x})$ and $K^+_($\mathbf{x}, \mathbf{x}')$ are the casual prior parameters. It is possible to improve CBO by considering DAG-GP$^+$ as surrogate model. For each $\mathbf{X}_s \in \mathcal{P}(\mathbf{X})$, instead of considering a single-task GP model as in Eq. 6, one can use $t_s(\mathbf{x}) \sim \mathcal{GP}(m_s(\mathbf{x}), K_s(\mathbf{x}, \mathbf{x}'))$ with $m_s(\mathbf{x})$ and $K_s(\mathbf{x}, \mathbf{x}')$ being the parameters computed as in Eqs. (4)-(5) in the main paper. This allows CBO to correctly place the next function evaluations thus significantly speeding up the convergence to the global optimum both with and without the causal prior.

# 7 Experiments

**Implementation details:** For all experiments, we assume Gaussian distributions for both the integrating measures and the conditional distributions in the DAGs. We optimise the parameters via maximum likelihood. We generate the observational data by sampling from the SCMs given below. In order to generate interventional data, we sample from a modified version of the SCM where the functional relationship corresponding to the intervened variable is substituted with a constant value. This is equivalent to sampling from the mutilated graph. We compute the integrals in Eqs. (4)–(5) via Monte-Carlo integration with 1000 samples. Finally, we fix the variance in the likelihood of Eq. (3) and fix the kernel hyper-parameters for both the RBF and causal kernel to standard values ($l = 1$, $\sigma_f^2 = 1$). More works need to be done to optimise these settings potentially leading to improved performances.

## 7.1 DAG1

***Do*-calculus derivations** For DAG1 (Fig. 1 (a)) we have $\mathbf{I} = \{Z\}$ and $\mathbf{C} = \varnothing$. The base function is thus given by $f = \mathbb{E}[Y|\text{do}\,(Z = z)]$. In this section we give the expressions for the functions in $\mathbf{T}$ and show each of them can be written as a transformation of $f$ with the corresponding integrating measure. Notice that in this case $f \in \mathbf{T}$.

$$
\begin{aligned}
\mathbb{E}[Y|\text{do}\,(X = x)] &= \int \mathbb{E}[Y|\text{do}\,(X = x), z]p(z|\text{do}\,(X = x))\mathrm{d}z \\
&= \int \mathbb{E}[Y|\text{do}\,(X = x), \text{do}\,(Z = z)]p(z|\text{do}\,(X = x))\mathrm{d}z \quad \text{by} \quad Y \perp\!\!\!\perp Z|X \text{ in } \mathcal{G}_{\overline{BXZ}} \\
&= \int \mathbb{E}[Y|\text{do}\,(Z = z)]p(z|\text{do}\,(X = x))\mathrm{d}z \quad \text{by} \quad Y \perp\!\!\!\perp X|Z \text{ in } \mathcal{G}_{\overline{XZ}} \\
&= \int f(z)p(z|\text{do}\,(X = x))\mathrm{d}z
\end{aligned}
$$

with $p(z|\text{do}\,(X = x)) = p(z|X = x)$.

$$
\mathbb{E}[Y|\text{do}\,(Z = z)] = f(z).
$$

$$
\mathbb{E}[Y|\text{do}\,(X = x), \text{do}\,(Z = z)] = \mathbb{E}[Y|\text{do}\,(Z = z)] = f(z)
$$
$$
\text{by} \quad Y \perp\!\!\!\perp X|Z \text{ in } \mathcal{G}_{\overline{XZ}}
$$

**SCM:**

$$
\begin{aligned}
X &= \epsilon_X \\
Z &= \exp(-X) + \epsilon_Z \\
Y &= \cos(Z) - \exp(-Z/20) + \epsilon_Y
\end{aligned}
$$

with $\epsilon_X \sim \mathcal{N}(0,1)$, $\epsilon_Z \sim \mathcal{N}(0,1)$ and $\epsilon_Y \sim \mathcal{N}(0,1)$. We consider the following interventional domains:

- $D(X) = [-5, 5]$
- $D(Z) = [-5, 20]$

## 7.2 DAG2

***Do*-calculus derivations** For DAG2 (Fig. 1 (b)) we consider $\{A, C\}$ to be non-manipulative. We have $\mathbf{I} = \{D, E\}$ and $\mathbf{C} = \{A, B\}$. The base function is thus given by $f = \mathbb{E}[Y|\text{do}\,(D = d), \text{do}\,(E = e), a, b]$. In this section we give the expressions for all the functions in $\mathbf{T}$ and show each of them can be written as a transformation of $f$ with the corresponding integrating measure.

**Intervention sets of size 1**

$$\mathbb{E}[Y|\text{do}\,(B=b)] = \int \mathbb{E}[Y|\text{do}\,(B=b)\,,d,e,a]p(d,e,a|\text{do}\,(B=b))\mathrm{d}d\mathrm{d}e\mathrm{d}a$$

$$= \int \mathbb{E}[Y|\text{do}\,(B=b)\,,\text{do}\,(D=d)\,,\text{do}\,(E=e)\,,a]p(d,e,a|\text{do}\,(B=b))\mathrm{d}d\mathrm{d}e\mathrm{d}a$$

$$\text{by}\quad Y\perp\!\!\!\perp D,E|B,A \text{ in } \mathcal{G}_{\overline{B}\underline{DE}}$$

$$= \int \mathbb{E}[Y|\text{do}\,(D=d)\,,\text{do}\,(E=e)\,,a]p(d,e,a|\text{do}\,(B=b))\mathrm{d}d\mathrm{d}e\mathrm{d}a \quad \text{by}\quad Y\perp\!\!\!\perp B|D,E,A \text{ in } \mathcal{G}_{\overline{BDE}}$$

$$= \int \mathbb{E}[Y|\text{do}\,(D=d)\,,\text{do}\,(E=e)\,,a,b']p(b')p(d,e,a|\text{do}\,(B=b))\mathrm{d}d\mathrm{d}e\mathrm{d}a\mathrm{d}b'$$

$$= \int f(d,e,a,b')p(b')p(d,e,a|\text{do}\,(B=b))\mathrm{d}d\mathrm{d}e\mathrm{d}a\mathrm{d}b'$$

with $p(b')p(d,e,a|\text{do}\,(B=b)) = p(b')p(a)p(d|e,a,B=b)p(e|a,B=b)$.

$$\mathbb{E}[Y|\text{do}\,(D=d)] = \int \mathbb{E}[Y|\text{do}\,(D=d)\,,e,a,b]p(a,b,e|\text{do}\,(D=d))\mathrm{d}a\mathrm{d}b\mathrm{d}e$$

$$= \int \mathbb{E}[Y|\text{do}\,(D=d)\,,\text{do}\,(E=e)\,,a,b]p(a,b,e|\text{do}\,(D=d))\mathrm{d}a\mathrm{d}b\mathrm{d}e \quad \text{by}\quad Y\perp\!\!\!\perp E|D,A,B \text{ in } \mathcal{G}_{\overline{D}\underline{E}}$$

$$= \int f(d,e,a,b)p(a,b,e|\text{do}\,(D=d))\mathrm{d}a\mathrm{d}b\mathrm{d}e$$

with $p(a,b,e|\text{do}\,(D=d)) = p(a)p(b)p(e|a,b)$.

$$\mathbb{E}[Y|\text{do}\,(E=e)] = \int \mathbb{E}[Y|\text{do}\,(E=e)\,,d,a,b]p(d,a,b|\text{do}\,(E=e))\mathrm{d}a\mathrm{d}b\mathrm{d}s$$

$$= \int \mathbb{E}[Y|\text{do}\,(E=e)\,,\text{do}\,(D=d)\,,a,b]p(d,a,b|\text{do}\,(E=e))\mathrm{d}a\mathrm{d}b\mathrm{d}d \quad \text{by}\quad Y\perp\!\!\!\perp D|E,A,B \text{ in } \mathcal{G}_{\overline{E}\underline{D}}$$

$$= \int f(d,e,a,b)p(d,a,b|\text{do}\,(E=e))\mathrm{d}a\mathrm{d}b\mathrm{d}d$$

with $p(d,a,b|\text{do}\,(E=e)) = p(a)p(b)p(d|b)$.

**Intervention sets of size 2**

$$\mathbb{E}[Y|\text{do}\,(B=b)\,,\text{do}\,(D=d)] = \int \mathbb{E}[Y|\text{do}\,(B=b)\,,\text{do}\,(D=d)\,,a,e]p(a,e|\text{do}\,(B=b)\,,\text{do}\,(D=d))\mathrm{d}a\mathrm{d}e$$

$$= \int \mathbb{E}[Y|\text{do}\,(B=b)\,,\text{do}\,(D=d)\,,a,\text{do}\,(E=e)]p(a,e|\text{do}\,(B=b)\,,\text{do}\,(D=d))\mathrm{d}a\mathrm{d}e$$

$$\text{by}\quad Y\perp\!\!\!\perp E|A,B,D \text{ in } \mathcal{G}_{\overline{BD}\underline{E}}$$

$$= \int \mathbb{E}[Y|\text{do}\,(D=d)\,,\text{do}\,(E=e)\,,a]p(a,e|\text{do}\,(B=b)\,,\text{do}\,(D=d))\mathrm{d}a\mathrm{d}e$$

$$\text{by}\quad Y\perp\!\!\!\perp B|A,D,E \text{ in } \mathcal{G}_{\overline{BDE}}$$

$$= \int \mathbb{E}[Y|\text{do}\,(D=d)\,,\text{do}\,(E=e)\,,a,b']p(b')p(a,e|\text{do}\,(B=b)\,,\text{do}\,(D=d))\mathrm{d}a\mathrm{d}b'\mathrm{d}e$$

with $p(b')p(a,e|\text{do}\,(B=b)\,,\text{do}\,(D=d)) = p(b')p(a)p(e|a,B=b)$.

$$\mathbb{E}[Y|\mathrm{do}\,(B=b)\,,\mathrm{do}\,(E=e)] = \int \mathbb{E}[Y|\mathrm{do}\,(B=b)\,,\mathrm{do}\,(E=e)\,,a,d]p(a,d|\mathrm{do}\,(B=b)\,,\mathrm{do}\,(E=e))\mathrm{d}a\mathrm{d}d$$

$$= \int \mathbb{E}[Y|\mathrm{do}\,(B=b)\,,\mathrm{do}\,(E=e)\,,a,\mathrm{do}\,(D=d)]p(a,d|\mathrm{do}\,(B=b)\,,\mathrm{do}\,(E=e))\mathrm{d}a\mathrm{d}d$$

$$\text{by}\quad Y\perp\!\!\!\perp D|A,B,E \text{ in } \mathcal{G}_{\overline{BE}\underline{D}}$$

$$= \int \mathbb{E}[Y|\mathrm{do}\,(D=d)\,,\mathrm{do}\,(E=e)\,,a]p(a,d|\mathrm{do}\,(B=b)\,,\mathrm{do}\,(E=e))\mathrm{d}a\mathrm{d}d$$

$$\text{by}\quad Y\perp\!\!\!\perp B|A,D,E \text{ in } \mathcal{G}_{\overline{BDE}}$$

$$= \int \mathbb{E}[Y|\mathrm{do}\,(D=d)\,,\mathrm{do}\,(E=e)\,,a,b']p(b')p(a,d|\mathrm{do}\,(B=b)\,,\mathrm{do}\,(E=e))\mathrm{d}a\mathrm{d}b'\mathrm{d}d$$

$$= \int f(d,e,a,b')p(b')p(a,d|\mathrm{do}\,(B=b)\,,\mathrm{do}\,(E=e))\mathrm{d}a\mathrm{d}b'\mathrm{d}d$$

with $p(b')p(a,d|\mathrm{do}\,(B=b)\,,\mathrm{do}\,(E=e)) = p(b')p(a)p(d|B=b)$.

$$\mathbb{E}[Y|\mathrm{do}\,(D=d)\,,\mathrm{do}\,(E=e)] = \int \mathbb{E}[Y|a,b,\mathrm{do}\,(D=d)\,,\mathrm{do}\,(E=e)]p(a,b|\mathrm{do}\,(D=d)\,,\mathrm{do}\,(E=e))\mathrm{d}a\mathrm{d}b$$

$$= \int f(d,e,a,b)p(a,b|\mathrm{do}\,(D=d)\,,\mathrm{do}\,(E=e))\mathrm{d}a\mathrm{d}b$$

with $p(a,b|\mathrm{do}\,(D=d)\,,\mathrm{do}\,(E=e)) = p(a)p(b)$.

**Intervention sets of size 3**

$$\mathbb{E}[Y|\mathrm{do}\,(B=b)\,,\mathrm{do}\,(D=d)\,,\mathrm{do}\,(E=e)] = \mathbb{E}[Y|\mathrm{do}\,(D=d)\,,\mathrm{do}\,(E=e)]$$
$$\text{by}\quad (Y\perp\!\!\!\perp B|D,E \text{ in } \mathcal{G}_{\overline{DEB}})$$

**SCM:**

$$U_1 = \epsilon_{YA}$$
$$U_2 = \epsilon_{YB}$$
$$A = U_1 + \epsilon_A$$
$$B = U_2 + \epsilon_B$$
$$C = \exp(-B) + \epsilon_C$$
$$D = \exp(-C)/10. + \epsilon_D$$
$$E = \cos(A) + C/10 + \epsilon_E$$
$$Y = \cos(D) + \sin(E) + U_1 + U_2 + \epsilon_y$$

with $\epsilon_i \sim \mathcal{N}(0,1), \quad \forall i \in \{YA, YB, A, B, C, D, E, y\}$. We consider the following interventional domains:

- $D(B) = [-3, 4]$
- $D(D) = [-3, 3]$
- $D(E) = [-3, 3]$

## 7.3 DAG3

***Do*-calculus derivations** For DAG3 (Fig. 1 (c)) we consider $\{\text{age}, \text{BMI}, \text{cancer}\}$ to be non-manipulative. We have $\mathbf{I} = \{\text{aspirin}, \text{statin}, \text{age}, \text{BMI}, \text{cancer}\}$ and $\mathbf{C} = \varnothing$. In this section we give the expressions for all the functions in $\mathbf{T}$ and show each of them can be written as a transformation of $f$ with the corresponding integrating measure.

$$\mathbb{E}[Y|\text{do}(\text{aspirin} = x)] = \int \cdots \int f(\text{aspirin}, \text{statin}, \text{age}, \text{BMI}, \text{cancer})$$
$$p(\text{statin}, \text{age}, \text{BMI}, \text{cancer}|\text{do}(\text{aspirin} = x))\text{dstatindagedBMIdcancer}$$

with $p(\text{statin}, \text{age}, \text{BMI}, \text{cancer}|\text{do}(\text{aspirin} = x)) = p(\text{cancer}|\text{age}, \text{BMI}, \text{aspirin}, \text{aspirin})p(\text{statin}|\text{age}, \text{BMI})p(\text{BMI}|\text{age})p(\text{age})$.

$$\mathbb{E}[Y|\text{do}(\text{statin} = x)] = \int \cdots \int f(\text{aspirin}, \text{statin}, \text{age}, \text{BMI}, \text{cancer})$$
$$p(\text{aspirin}, \text{age}, \text{BMI}, \text{cancer}|\text{do}(\text{statin} = x))\text{daspirindagedBMIdcancer}$$

with $p(\text{aspirin}, \text{age}, \text{BMI}, \text{cancer}|\text{do}(\text{statin} = x)) = p(\text{cancer}|\text{age}, \text{BMI}, \text{aspirin}, \text{aspirin})p(\text{aspirin}|\text{age}, \text{BMI})p(\text{BMI}|\text{age})p(\text{age})$.

$$\mathbb{E}[Y|\text{do}(\text{aspirin} = x), \text{do}(\text{statin} = z)] = = \int \cdots \int f(\text{aspirin}, \text{statin}, \text{age}, \text{BMI}, \text{cancer})$$
$$p(\text{age}, \text{BMI}, \text{cancer}|\text{do}(\text{aspirin} = x), \text{do}(\text{statin} = z))\text{dagedBMIdcancer}$$

with $p(\text{age}, \text{BMI}, \text{cancer}|\text{do}(\text{aspirin} = x), \text{do}(\text{statin} = z)) = p(\text{cancer}|\text{age}, \text{BMI}, \text{aspirin}, \text{aspirin})p(\text{BMI}|\text{age})p(\text{age})$.

**SCM:**

$$\text{age} = \mathcal{U}(55, 75)$$
$$\text{bmi} = \mathcal{N}(27.0 - 0.01 \times \text{age}, 0.7)$$
$$\text{aspirin} = \sigma(-8.0 + 0.10 \times \text{age} + 0.03 \times \text{bmi})$$
$$\text{statin} = \sigma(-13.0 + 0.10 \times \text{age} + 0.20 \times \text{bmi})$$
$$\text{cancer} = \sigma(2.2 - 0.05 \times \text{age} + 0.01 \times \text{bmi} - 0.04 \times \text{statin} + 0.02 \times \text{aspirin})$$
$$Y = \mathcal{N}(6.8 + 0.04 \times \text{age} - 0.15 \times \text{bmi} - 0.60 \times \text{statin} + 0.55 \times \text{aspirin} + 1.00 \times \text{cancer}, 0.4)$$

We consider the following interventional domains:

- $D(\text{aspirin}) = [0, 1]$
- $D(\text{statin}) = [0, 1]$

### 7.4 Extended version of Fig. 2

Fig. 2 shows how DAG-GP captures the behaviour of both intervention functions in areas where $\mathcal{D}^I$ is not available (see area around $x = -2$) while reducing the uncertainty via transfer. As the two interventional ranges differ, the interventional points for both $X$ and $Z$ are in different parts of the input space. Specifically, $\mathcal{D}_Z^I$ contains points that are outside of $D(X)$. However, the non linear mapping between the two functions allows to transfer information in this scenario. Notice that, due to the non linear mapping, an observation for $Z$, say at $z = 10$, will *not* correspond to an improved posterior estimation for $X$ at $x = 10$ as $z = 10$ will be mapped to another part of $D(X)$. At the same time, the function $t_X(\mathbf{x})$ will not pass through the points in $\mathcal{D}_Z^I$ as it is the case in standard GP regression. Finally, notice how the orange lines in the left panels give the do-calculus results (labelled as "do") which are used to construct the causal prior for the DAG-GP$^+$ in the right panels.

### 7.5 Additional experimental results

Here we give additional experimental results for both the synthetic examples and the health-care application. Tab. 1 gives the fitting performances, across intervention functions and replicates, when $N = 500$.

Figure 2: Posterior mean and variance for $t_X(\mathbf{x})$ and $t_Z(\mathbf{z})$ in the DAG of Fig. 4 (a) (without the red edge). For both plots $m_i(\cdot)$ and $K_i(\cdot, \cdot)$ give the posterior mean and standard deviation respectively.

Table 1: RMSE with $N = 500$

|  | DAG-GP$^+$ | DAG-GP | GP$^+$ | GP | *do*-calculus |
|---|---|---|---|---|---|
| DAG1 | **0.48** | 0.57 | 0.60 | 0.77 | 0.55 |
|  | (0.07) | (0.08) | (0.15) | (0.27) | - |
| DAG3 | 0.50 | **0.42** | 0.58 | 1.26 | 2.87 |
|  | (0.11) | (0.13) | (0.10) | (0.11) | - |
| DAG4 | **0.09** | 0.44 | 0.54 | 0.89 | 0.22 |
|  | (0.05) | (0.12) | (0.08) | (0.23) | - |