[Reviews · NeurIPS 2020]

Review 1

Summary and Contributions: This paper presents a new method for learning correlation among multiple intervention functions using multi-task GP. This enables the causal transfer across intervening experiments with different sets of variables.

Strengths: The causal transfer task that this paper aims to address is interesting and also practically useful in various domains such as healthcare & operation research that requires sequential intervening experiments (e.g., intervening experimentation might be necessary to determine accurately the averaged effect of a treatment that might be confounded with other latent factors if we only look at observational data) This paper introduces an interesting perspective on how multi-task GP can be used to learn the correlation across intervention experiments (modeled as random functions), which can be used as quantitative measure to assess the value of causal information that one would gain from an experiment given the interventional data gathered from previous ones. I have not seen multi-task GP being adapted in such context before and the idea sounds very new and refreshing to me. In particular, it bridges very well between the two seemingly different aspects of causal learning: experimental design & transferability. I can imagine that the idea presented here would be very practical when people have a budget on how many experiment they can afford, which would render asymptotic sequential decision making methods that only guarantee optimality in the limit (and in worst-case scenarios) less useful.

Weaknesses: Although the high-level idea and the tech detail that substantiate it makes sense, the complexity analysis of the proposed algorithm was not provided. I think such analysis is necessary to thoroughly demonstrate the method's practicality -- please discuss this in the rebuttal. The paper assumes a directed graph relating the variables is known. In practice, how practical is this assumption? In fact, different graphs might induce the same marginal distribution of data (so we are not 100% sure if the provided graph is the correct one). Wouldn't it be a problem if we want to learn the causal relationship (not their correlation) among variables?

Correctness: I have made high-level check at all technical derivations & all make sense to me. The empirical evaluation is also sufficient and well-demonstrated.

Clarity: In general, I enjoy reading this paper. It is well-written but I have to say that the notations are a bit too dense, especially when multiple notations were used to indicate the same thing on multiple occasions. I understand this is to make the math more concise but at the same time, it also makes it hard for readers to follow the authors' thought process.

Relation to Prior Work: From a laymen perspective (on causality), I think this paper makes a good contribution. But of course, my expertise does not span the area of causality so I cannot speak definitively for its novelty in the context of causality -- I would delegate this to the other experts. I am, however, confident to say that I have not seen any similar causal work in the multi-task GP context before & I think this work is a great add to the existing GP literature.

Reproducibility: Yes

Additional Feedback: -- post-rebuttal feedback -- I have read your rebuttal.


Review 2

Summary and Contributions: The manuscript proposes DAG-GP the first method that allows sharing information across continuous interventions and experiments to learn the causal effects of interventions on different subsets of variables. DAG-GP infers correlations between functions in different input spaces. Results demonstrate when and how DAG-GP can be constructed depending on the topology of the DAG. Experiments investigate inferences and uncertainty calibration, as well as optimal intervention for real and synthetic data.

Strengths: The strengths of the paper are the importance of the problem of causal learning and the integration of GPs inspired by multitask learning applied to the problem of modeling intervention functions for sharing information across experiments is interesting. The theoretical analysis describing when DAG-GP can be constructed is nice. Results from a variety of experiments are provided to compare the performance of DAG-GP to alternative approaches on synthetic and real data.

Weaknesses: A weakness is that it could be more concise and precise in convincingly demonstrating the novelty. One example is the experiments, which do not appear to compare with alternative methods discussed in the related work. Of course, it is not fun to try to implement previous work if it is not already available (sometimes even if it is), but it is important to provide a precise, quantitative sense of when and why this method should outperform other existing methods. The current results focus on comparing against some more standard baselines. The results also focus on a few examples that do not seem to highlight when, if ever, one would not want to use this approach. Indeed, section 5.1 suggests that DAG-GP+ is a robust choice for any application, which is a strong conclusion from rather limited evidence and analysis. Relatedly, various aspects of the paper were not self-contained. For example, the “causal prior” was used, without, as far as I could tell, a definition, yet is featured throughout the results. [On my third read, I found the definition on line 189. Given the centrality of this prior to the results, why was it not discussed in related work?] Similarly, figure 2, which is meant to illustrate the advantages of DAG-GP could be clearer. For example, it is stated that the availability of data for z contributes to decreased uncertainty, but I do not see any reference to what the data are and how they help in this way. In the right panel, the comparison between DAG-GP and DAG-GP+ would be aided by providing details about the causal prior used in DAG-GP+. Another example is the active learning and Bayesian optimization demonstrations which use algorithms that are not described beyond a reference. This makes an already dense paper harder to read. The active learning and Bayesian optimization results are interesting, but I would have rather seen a more comprehensive analysis of the strengths of DAG-GP relative to prior work and limitations of the approach. To summarize, there are considerable merits to the paper. However, the current exposition makes it challenging to accurately analyze the advance of the approach. For me, the paper would be stronger if there was greater focus on understanding DAG-GP itself, demonstrating when, why and how much it improves performance, and ensuring that the paper includes all details relevant to understanding and evaluating the results.

Correctness: The results appear complete and correct. At the conceptual/structure level, I believe the paper could be more clear. At the sentence level the paper is clear.

Clarity: At the conceptual/structure level, I believe the paper could be more clear. At the sentence level the paper is clear.

Relation to Prior Work: The relation to prior work is discussed. There are ways in which the relationship to prior work could be improved including clarifying the relationship between the causal prior from prior work and DAG-GP proposed here, and explicit comparison and quantification of the contribution of DAG-GP versus prior work.

Reproducibility: Yes

Additional Feedback: I was unsure what the authors were referring to by “in the sequel”. This appears at several points. I do not know what sequel they are referring to. - Further explanation in the caption of figure 4 would be helpful. The current version makes no reference to the three panels and what they uniquely contribute. ====Post response======= After reading the review and discussion with the other reviewers, my rating is unchanged.


Review 3

Summary and Contributions: The paper proposes a technique for combining evidence from multiple experimental datasets into a Gaussian process model for estimating causal effects. Specifically, the graph structure induces a weighting (an integrating measure) on the covariance between instances from different experiments.

Strengths: The ideas presented in this paper are compelling, appear to be sound, and are significant. Specifically, the idea of using the graph structure to weight the covariance between measurements from multiple experiments is clever and appears to be effective.

Weaknesses: The paper would be significantly improved by emphasizing the intuition behind the main results. For example, how does the graph structure relate to the integrating measure? A simple example with a visual representation of the induced covariance kernel would help provide this intuition. Given the space constraints of a NeurIPS submission, the paper would likely be more effective without the Bayesian Optimization results. As written, the submission does not provide enough detail about how DAG-GP should be modified to incorporate into a BO procedure. The authors could use this additional space to provide more clarity on the main results. See below for additional detailed comments.

Correctness: The claims and methods appear to be correct, although the description of the experiments is too sparse to verify this. Specifically, the authors should clarify which variables are included in the target causal estimand, and which variables were intervened upon in the experimental datasets. In addition, the authors describe one of the main benefits of DAG-GP is its ability to provide estimates of uncertainty in the effects of an intervention. The authors should evaluate this claim by reporting not just how accurate the mean estimate is, but how much posterior density the model places on the true causal estimated.

Clarity: The paper is generally well written, although there are some minor errors that inhibit its clarity. As discussed above, more emphasis should be placed on providing intuition of the main results.

Relation to Prior Work: The differences and connections to most prior work are discussed in this submission. However, the authors should clearly discuss how the submission relates to “General Identifiability with Arbitrary Surrogate Experiments” (Lee et al, UAI 2019). In addition to a discussion in the related work section, the authors should compare empirically against a baseline that (1) applies the gID procedure to construct an an expression for the target estimated in terms of observational and experimental distributions and (2) estimate the observational and experimental distributions from available data. This should be repeated twice, once with a GP-based estimator for (2) and once with some other kind of nonparametric estimator.

Reproducibility: No

Additional Feedback: Line 18: “Solving decision making problems in a variety of domains such as healthcare, systems biology or operations research, requires experimentation.” -> This statement ignores the large body of research on observational causal inference. Experiments are certainly useful in many settings, but they’re not always required. This statement should be softened. Line 49: “In do-calculus, different intervention functions are modeled individually and there is no information shared across experiments” -> This statement is incorrect. The do-calculus provides a set of rules for translating between expressions containing the do-operator to expressions that only contain observable marginal and conditional distributions. Information is shared between interventions that, when translated to observational expressions, contain the same conditional or marginal distributions. A thorough editorial review of the paper would significantly improve its clarity. See below for a representative, but not comprehensive, set of suggested corrections. Line 23: “once one of them is knockout” -> “once one of them is knocked out”. Line 41: “thus allowing to define” -> “thus allowing the definition of” Line 48: “the do-calculus allows to predict” -> “the do-calculus enables prediction” Line 106: “do-calculus allows to estimate” -> “do-calculus allows the estimation of” Line 133: “Next results” -> “The following results” POST AUTHOR RESPONSE: I have read the authors' response and the other reviews. As a result, I have increased my score from a 5 to a 6 to reflect the fact that the procedure the authors propose lends itself to experimental design/active learning more naturally than does the gID procedure. That being said, I still maintain that the authors should add the gID baseline I previously recommended. In their response, the authors state the following: "In our paper we assume full identifiability of the causal effects (line 108). When this is the case, [1] reduces to [2] and p(Y |do(X = x)) is computed from observational data via do-calculus ∀X." To paraphrase, my understanding is that the authors are claiming that if a causal estimand is nonparametrically identifiable from observational data alone, the gID procedure (Algorithm 1 in Lee et al, UAI 2019) will only rely on observational data, and won't take advantage of any experimental data. This is incorrect. For example, consider a three variable causal graph where an observed confounder U has a directed edge to treatment X and outcome Y. In this example, we want to estimate P(Y|do(X=x)) for some x. If we have an experimental dataset where we intervened on X (which includes x in the support), gID will terminate at line 2 returning only the results of that experiment. We certainly could estimate P(Y|do(X=x)) from observational data using the backdoor adjustment, but this isn't what gID does in this simple scenario despite the authors' claim. It's very likely that the methods presented in this paper would outperform the gID baseline in this simple scenario, especially if this experimental dataset was small relative to observational or other available experimental datasets, however this is a claim that should be evaluated empirically.


Review 4

Summary and Contributions: This paper proposes the first multi-task causal Gaussian process (GP) modeling framework, name as DAG-GP, which enables information sharing across interventions and experiments on different variables. The paper first provides theoretical foundations of the model. The paper then provides synthetic/real-world experiments to demonstrate the effectiveness of DAG-GP on various important problems, such as (1) learning across interventions and experiments on different variables (2) active learning (3) Bayesian optimization.

Strengths: I believe the paper proposed a promising new framework for learning across experiments. The paper is well-written, and provides a solid theoretical foundation. Furthermore, the paper shows the effectiveness of the model on various areas with synthetic/real-world experiments.

Weaknesses: 1. I believe this paper could provide even more insight if it includes a brief discussion on single task models, especially because in the experiment section DAG-GP is compared against them. 2. Is it possible to provide more details on how the synthetic data are generated, potentially in supplementary?

Correctness: As far as my knowledge and understanding goes, I believe the claim and empirical methodology are correct. However, as mentioned in the "weakness" area, it would be great if the paper could share more detail on how the synthetic data are generated.

Clarity: Yes, the paper is well-written.

Relation to Prior Work: To the best of my knowledge on previous contributions, the paper has clearly discussed how this work differs from previous contributions.

Reproducibility: No

Additional Feedback: Comments on reproducibility: The paper provides some good detail on the experiments, but as mentioned earlier, to enable full reproducibility, it would be great if the paper could share even more detail on how the synthetic data are generated.

[Author Response · NeurIPS 2020]

We thank the reviewers for their detailed and helpful feedback. We are glad that *all* reviewers agree on the novelty of
the paper and recognise its original and significant contribution to the machine learning community. The reviewers
acknowledge that the ideas presented in the paper are compelling, sound and appear to be effective (R3), offering a
great add to the GP literature (R1) which is also supported by a solid and an interesting theoretical foundation (R2,
R4). In the following discussion, we address three main points: comparison to related work, further intuition behind
the DAG-GP model and additional experimental details. We respond point-by-point to the comments raised by each
reviewer. We hope our detailed response below will further highlight the paper's quality and originality.

**R#1:** *(1) Knowledge of the causal graph.* In this paper we indeed assume the DAG to be known and we mention
the integration of the DAG-GP model with a causal discovery algorithm as a future direction. Different causal
discovery algorithms have been proposed which could be used as a pre-processing step. In addition, one could use the
interventional data to discriminate across graphs within the Markov equivalence class. Analysing what happens when
the DAG is unknown goes beyond the scope of this paper and remains an area of future research. We have clarified this
point. *(2) Complexity analysis.* The time complexity of the algorithm is $\mathcal{O}(N^3)$ with $N$ denoting the size of $\mathcal{D}^I$. This
complexity can be reduced by resorting to sparse GP approximations e.g. inducing points approximations.

**R#2:** *(1) Comparison to related work.* As highlighted by both R#1 and R#4, this approach is completely new in the
GP literature. Existing multi-output GP models are not applicable to our setting (see line 79-83) and are thus not
comparable to the DAG-GP model. In the causality literature, studies have focused on observational causal inference
and transferability problems where the goal is to transfer the causal effects of *one* given variable across environments.
On the contrary, our goal is to transfer information *across all* causal effects in a single environment thus existing studies
are not comparable. We have further clarified this point in Section 1.2. We have also mentioned the causal prior in
the related work and highlighted its definition in Section 3.2. See R#3 (1) for more details on relationship to existing
works. *(2) Intuition behind the DAG-GP and experimental results.* We have used the additional page to provide a deeper
analysis of the DAG-GP model and additional implementation details to aid understanding of the results. In particular,
as suggested by R#3, we have shown how different covariance structures are linked to different DAGs and capture
different levels of transfer within tasks. We have also added a brief intro to BO and AL and clarified how the DAG-GP
can be used as a surrogate model within these two frameworks. *(3) Additional comments.* We have clarified Fig 4 and
Fig 2. We have added an extended version of Fig 2 in the appendix where we have plotted the prior mean functions
used by DAG-GP+, the posterior distributions obtained by the alternative models and the $Z$ interventional data.

**R#3:** *(1) Comparison to related work.* We have added [1] to the related work and discussed how its goal and setting
differ from our paper making it not comparable. In our paper we assume full identifiability of the causal effects (line
108). When this is the case, [1] reduces to [2] and $p(\boldsymbol{Y}|do(\boldsymbol{X}=\boldsymbol{x}))$ is computed from observational data via do-calculus
$\forall \boldsymbol{X}$. This option is included as a benchmark. When $\exists \boldsymbol{X}$ s.t. $p(\boldsymbol{Y}|do(\boldsymbol{X}=\boldsymbol{x}))$ is not identifiable and interventional
data for $\boldsymbol{X}$ is not available, one need to resort to gID[1] or zID[3] to find an expression in terms of the available
data/distributions. Our approach cannot deal with these settings as the integrating measures wouldn't be identifiable thus
preventing the propagation of $p(f)$. Finally, notice that [1] aims at expressing the causal effects of *one* given variable in
terms of the available distributions. While one can repeat the procedure for all possible intervention sets in $\mathcal{G}$, [1] is not
gonna necessarily express all causal effects via a shared interventional distribution. The goal of our analysis is exactly
that: identifying a *shared interventional representation* while providing a probabilistic model to transfer information in
practice. Thanks for highlighting this connection. We will repeat the experiments using a kernel density estimator for
the observational distributions so as to show consistency of the results across different non-parametric methods. See
R#2 (1) above for more details on relationship to existing works. *(2) Intuition behind the DAG-GP and experimental
results.* As suggested, we have shown how different covariance structures are linked to different DAGs and capture
different levels of transfer within tasks. We have used the additional page to provide more experimental details and a
brief introduction to BO and AL clarifying how the DAG-GP model can be used as a surrogate model. *(3) Additional
comments.* Line 18: We have softened this statement. Notice however, that observational causal inference methods (and
the do-calculus) provide accurate estimates of the causal effects for interventional values that are observed in the data
and for which the observational distributions are accurately estimated. For interventional values not observed in the
data one needs to resort to experiments. Line 49: While our results are indeed based on the do-calculus, applying these
rules is not enough to transfer information. The do-calculus does not provide a way to obtain expressions in terms of
the same *shared* base function which is our main contribution.

**R#4:** We have used the additional page to expand the discussion on single-task models in Section 4. *(1) Experimental
details.* As mentioned above (see R#2 (2) and R#3 (2)) we will add further implementation details and we will provide
the code base to ensure full reproducibility. We have expanded Section 5 in the supplement so as to further clarify the
data generating mechanism for the synthetic data.

[1] Lee, Sanghack, Juan D. Correa, and Elias Bareinboim. "General Identifiability with Arbitrary Surrogate Experiments." UAI 2019.
[2] Tian, Jin, and Judea Pearl. "A general identification condition for causal effects." Aaai/iaai 2002. [3] Bareinboim, Elias, and
Judea Pearl. "Causal inference by surrogate experiments: z-identifiability." UAI 2012.


[Meta-Review · NeurIPS 2020]

The paper is sound and makes a significant contribution. The experiments demonstrate the effectiveness of the method. The paper is well written, but some technical aspects of the paper were not self-contained and definitions are missing. While minor issues remain after the author response, the reviewers agree that the paper is above the acceptance threshold.